# Hemodynamics regulate spatiotemporal artery muscularization in the developing circle of Willis

Siyuan Cheng[1,2,3†], Ivan Fan Xia[1,2,3†], Renate Wanner[1,2,3], Javier Abello[4], Amber N Stratman[4], Stefania Nicoli[1,2,3]*

[1]Department of Genetics, Yale School of Medicine, New Haven, United States; [2]Yale Cardiovascular Research Center, Section of Cardiology, Department of Internal Medicine, Yale School of Medicine, New Haven, United States; [3]Vascular Biology & Therapeutics Program, Yale School of Medicine, New Haven, United States; [4]Department of Cell Biology & Physiology, School of Medicine, Washington University in St. Louis, St. Louis, United States

*For correspondence:
stefania.nicoli@yale.edu

†These authors contributed equally to this work

Competing interest: The authors declare that no competing interests exist.

**Abstract** Vascular smooth muscle cells (VSMCs) envelop vertebrate brain arteries and play a crucial role in regulating cerebral blood flow and neurovascular coupling. The dedifferentiation of VSMCs is implicated in cerebrovascular disease and neurodegeneration. Despite its importance, the process of VSMC differentiation on brain arteries during development remains inadequately characterized. Understanding this process could aid in reprogramming and regenerating dedifferentiated VSMCs in cerebrovascular diseases. In this study, we investigated VSMC differentiation on zebrafish circle of Willis (CoW), comprising major arteries that supply blood to the vertebrate brain. We observed that arterial specification of CoW endothelial cells (ECs) occurs after their migration from cranial venous plexus to form CoW arteries. Subsequently, *acta2*+ VSMCs differentiate from *pdgfrb*+ mural cell progenitors after they were recruited to CoW arteries. The progression of VSMC differentiation exhibits a spatiotemporal pattern, advancing from anterior to posterior CoW arteries. Analysis of blood flow suggests that earlier VSMC differentiation in anterior CoW arteries correlates with higher red blood cell velocity and wall shear stress. Furthermore, pulsatile flow induces differentiation of human brain PDGFRB+ mural cells into VSMCs, and blood flow is required for VSMC differentiation on zebrafish CoW arteries. Consistently, flow-responsive transcription factor *klf2a* is activated in ECs of CoW arteries prior to VSMC differentiation, and *klf2a* knockdown delays VSMC differentiation on anterior CoW arteries. In summary, our findings highlight blood flow activation of endothelial *klf2a* as a mechanism regulating initial VSMC differentiation on vertebrate brain arteries.

## eLife assessment

This study provides the first analysis of vascular stabilization on the critical and evolutionarily conserved structure around the Circle of Willis in the brain, strengthened by using parallel in vivo and in vitro experimental approaches. The evidence supporting the claims is **solid** and the work will be **valuable** for scientists studying developmental and disease-related vascular stabilization.

## Introduction

Vascular smooth muscle cells (VSMCs) are contractile mural cells wrapping around endothelial cells (ECs) of large vessels, especially arteries (*Ando et al., 2022*; *Donadon and Santoro, 2021*; *Stratman et al., 2020*). The expression of contractile proteins, such as alpha-smooth muscle actin

(encoded by *acta2*), distinguishes VSMCs from pericytes, the other type of mural cells primarily associated with small vessels (*Bahrami and Childs, 2020*; *Donadon and Santoro, 2021*; *Stratman et al., 2020*). VSMCs are important for arterial homeostasis (*Basatemur et al., 2019*). In the brain, VSMC constriction and relaxation are essential for functional hyperemia in neurovascular coupling (*Hill et al., 2015*; *Kaplan et al., 2020*). Phenotype switching or dedifferentiation of VSMCs, marked by lower expression of contractile proteins, is prominent in pathological progression of atherosclerosis (*Basatemur et al., 2019*). VSMC phenotype switching or dedifferentiation also contributes to cerebrovascular disease and neurodegeneration (*Aguilar-Pineda et al., 2021*; *Chou et al., 2022*; *Frösen and Joutel, 2018*; *Milewicz et al., 2010*; *Oka et al., 2020*; *Poittevin et al., 2014*). As development and regeneration share many common molecular and cellular mechanisms (*Nowotarski and Sánchez Alvarado, 2016*), insights into VSMC differentiation on brain arteries during development may inform strategies to reprogram and regenerate dedifferentiated VSMCs and alleviate cerebrovascular disease.

VSMCs differentiate from heterogenous perivascular progenitors of mesoderm and neural crest origins (*Ando et al., 2019*; *Donadon and Santoro, 2021*; *Whitesell et al., 2019*). Previous research described various mechanisms that regulate VSMC differentiation. Autonomous Notch activation is required for specification of *pdgfrb+* mural cell progenitors from peri-arterial mesenchyme, and these progenitors later differentiate into *acta2+* VSMCs (*Ando et al., 2019*). Arterial Notch signaling activated by blood flow is necessary for *acta2+* VSMC emergence on zebrafish trunk dorsal aorta (*Chen et al., 2017*). Chemokine signaling promotes VSMC association with zebrafish dorsal aorta, whereas blood flow-responsive transcription factor krüppel-like factor 2 (encoded by *klf2a* in zebrafish) prevents their association with adjacent cardinal vein (*Stratman et al., 2020*). Endothelial BMP signaling and autonomous *foxc1* expression regulate VSMC differentiation in zebrafish ventral head (*Watterston et al., 2019*; *Whitesell et al., 2019*). Autonomous expression of an ATP-sensitive potassium channel modulates VSMC differentiation from *pdgfrb+* progenitors in the brain (*Ando et al., 2022*). These studies suggest that VSMC differentiation is highly organotypic.

The circle of Willis (CoW) consists of major arteries that supply blood to the vertebrate brain, including internal carotid arteries (ICAs) and posterior communicating arteries (PCAs) (*Campbell et al., 2019*; *Schröder et al., 2020*). In early development, ICAs are the first large arteries to form and supply all circulation in the developing brain (*Kathuria et al., 2011*; *Menshawi et al., 2015*). In adults, posterior circulation becomes independent, and ICAs remain major feeding arteries in anterior, while connected to posterior circulation with PCAs, completing a circle of arteries (*Kathuria et al., 2011*; *Schröder et al., 2020*). In adults, intracranial aneurysms are frequently found close to CoW arteries (*Frösen and Joutel, 2018*). CoW arteries are wrapped by VSMCs, and VSMC dedifferentiation and hyperplasia are described in carotid atherosclerosis and Moyamoya disease (*Chou et al., 2022*; *Fox et al., 2021*). Spatiotemporal dynamics of VSMC differentiation on vertebrate CoW arteries during development have not been thoroughly investigated. VSMC differentiation is difficult to observe *in vivo* with mammalian models, as mammalian embryos develop *in utero* and depend on maternal circulation. Zebrafish is an important vertebrate model for vascular development, as zebrafish embryos are externally fertilized, optically clear, and thus accessible to confocal live imaging of developing blood vessels (*Stratman and Weinstein, 2021*). Establishment of zebrafish fluorescent transgenic reporter lines with *pdgfrb* and *acta2* promoters enables *in vivo* visualization and assessment of mural cells and VSMCs (*Ando et al., 2016*; *Whitesell et al., 2014*). In addition, zebrafish CoW shows comparable geometry to human CoW: the caudal division of the internal carotid artery (CaDI) in zebrafish is comparable to human ICA, and the posterior communicating segment (PCS) in zebrafish is comparable to human PCA (*Isogai et al., 2001*). Taking advantage of the zebrafish model, we describe the spatiotemporal pattern of VSMC differentiation on CoW arteries, and associate this pattern with blood flow activation of *klf2a* in arterial ECs.

**Zebrafish CoW vessel nomenclature**

| | |
|---|---|
| CaDI | Caudal division of the internal carotid artery |
| BCA | Basal communicating artery |
| PCS | Posterior communicating segment |

## Results

### Artery muscularization is spatiotemporally regulated in CoW arteries

In zebrafish brain, arteries are formed by ECs from cranial venous plexus (*Fujita et al., 2011*). Recent single-cell transcriptome profiling of vascular cells in prenatal human brain also supports an endothelial differentiation trajectory from proliferative venous ECs to quiescent arterial ECs (*Crouch et al., 2022*). To establish the temporal sequence of CoW arterial specification and VSMC differentiation, we started with confocal live imaging of fluorescent transgenic reporter lines Tg(*flt4:yfp*)[hu4881], which labels primitive venous ECs, and Tg(*kdrl:hras-mcherry*)[s896], which is expressed in all ECs but enriched in arterial ECs, at four stages starting at 32 hours post fertilization (hpf), when CoW arteries are assembled (*Chi et al., 2008*; *Hogan et al., 2009*; *Isogai et al., 2001*). As arterial ECs express higher *kdrl*, we used the intensity of *mcherry* as an indicator of artery specification (*Figure 1*; *Figure 1—source data 1*; *Chi et al., 2008*). We found that when ECs from cranial venous plexus migrated to form CoW arteries at 32 hpf, they still retained primitive venous identity, as indicated by *flt4* expression (*Figure 1A*). When connections between cranial venous plexus and CoW arteries disappeared at 54 hpf, CoW arteries no longer expressed primitive venous *flt4* (*Figure 1B*). Expression of arterial enriched *kdrl* in CaDI and PCS at 3 days post fertilization (dpf) was significantly higher than 32 hpf (*Figure 1A, C, and E*). At 4 dpf, *kdrl* expression in CoW arteries remained comparable to 3 dpf, suggesting that arterial specification of CoW was completed by 3 dpf (*Figure 1C–E*). Together, these data suggest CoW morphogenesis precedes arterial specification.

Previous research suggested that *acta2*+ VSMCs on CoW arteries differentiate from *pdgfrb*+ mural cell progenitors (*Ando et al., 2022*; *Ando et al., 2019*). Thus, to characterize spatiotemporal dynamics of VSMC differentiation on CoW arteries, we performed confocal live imaging of Tg(*acta2:mcherry*)[ca8], TgBAC(*pdgfrb:egfp*)[ncv22], and Tg(*kdrl:cerulean*)[sd24], to visualize respectively VSMCs (red), *pdgfrb*+ progenitors (green), and developing arteries (white) at four stages starting from 32 hpf (*Figure 2*; *Ando et al., 2016*; *Page et al., 2013*; *Whitesell et al., 2014*). We found that *pdgfrb*+ progenitors emerged on CoW arteries around 54 hpf after CoW morphogenesis, and most *pdgfrb*+ progenitors did not express *acta2* at 54 hpf (*Figure 2A, B, E, and F*, and *Figure 2—figure supplement 1A, B*, *Figure 2—source data 1*). The number of *pdgfrb*+ progenitors on CoW arteries increased temporally from 54 hpf to 3 dpf (*Figure 2B, C, and E*). Notably, while many *pdgfrb*+ cells on CaDI differentiated into *acta2*+ VSMCs by 3 dpf, *acta2*+ VSMCs were hardly observed on adjacent BCA or PCS, suggesting that VSMC differentiation started between 54 hpf and 3 dpf on CaDI but not BCA or PCS (*Figure 2B, C, E, and F*, and *Figure 2—figure supplement 1A, B*, *Figure 2—source data 1*). At 4 dpf, the number of *pdgfrb*+ cells on CoW arteries remained comparable to 3 dpf, while the number of *acta2*+ VSMCs on CaDI increased (*Figure 2C–F* and *Figure 2—figure supplement 1A, B*). Importantly, most *pdgfrb*+ cells on BCA and PCS only started to express *acta2* around 4 dpf, much later than CaDI (*Figure 2C–F* and *Figure 2—figure supplement 1A, B*). These findings support that differentiation of *acta2*+ VSMCs from *pdgfrb*+ progenitors progresses spatially and temporally along the anterior to posterior direction on CoW arteries.

### CoW arteries have spatiotemporal difference in hemodynamics

Our data suggest that VSMCs on CaDI of anterior CoW differentiate much earlier than BCA and PCS of posterior CoW, although all *acta2*+ VSMCs on CoW differentiated from *pdgfrb*+ progenitors after they were recruited to CoW (*Figure 2* and *Figure 2—figure supplement 1*). In addition, anterior and posterior CoW vascular morphogenesis and arterial specification occurred around the same time (*Figure 1*). We speculate that hemodynamic distribution may contribute to the spatiotemporal difference in VSMC differentiation on CoW arteries. CaDI receive proximal arterial feed through lateral dorsal aorta from cardiac outflow tract (*Isogai et al., 2001*). Furthermore, CaDI differs from BCA and PCS in terms of overall vascular geometry, which is among determinants of local vascular hemodynamic forces such as wall shear stress (WSS) (*Katritsis et al., 2007*). To explore this possibility, we performed microangiography with a high molecular weight fluorescent dextran to determine each CoW artery diameter. We found that there was no significant difference among CaDI, BCA, and PCS (*Figure 3A–D*, *Figure 3—source data 1*). Then, we measured red blood cell (RBC) velocity with axial line scanning microscopy to analyze Tg(*kdrl:gfp;gata1:DsRed*)[zn1/sd2] zebrafish at 54 hpf, 3 dpf, and 4 dpf (*Barak et al., 2021*). The RBC velocity across CoW arteries was differentially distributed, with CaDI having the highest RBC velocity from 54 hpf to 4 dpf (*Figure 3E–G*, *Figure 3—source data 1*). This

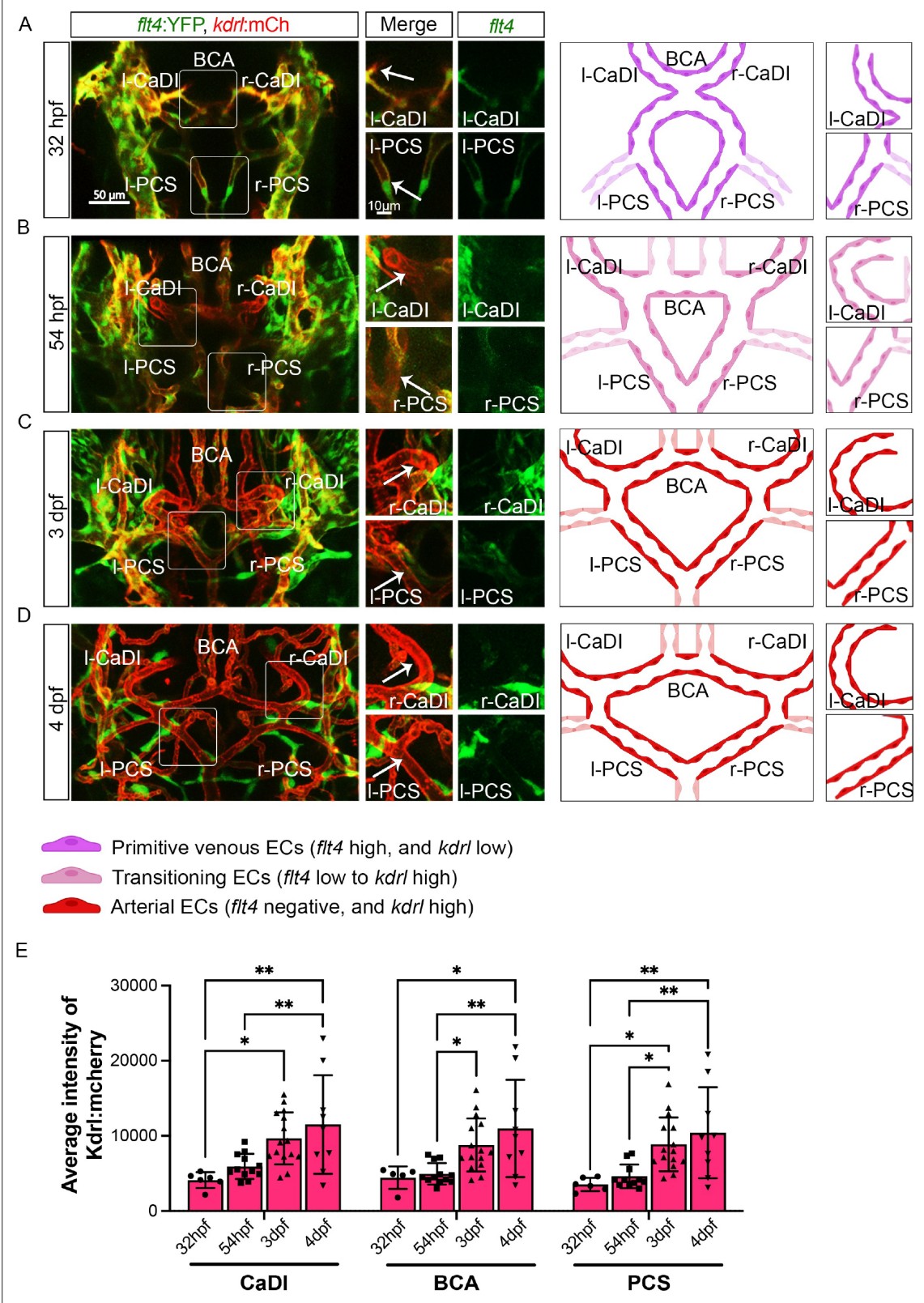

**Figure 1.** Arterial specification of endothelial cells (ECs) in circle of Willis (CoW) arteries. (**A–D**) Confocal live images of CoW arteries in Tg(*flt4:yfp*, *kdrl:hras-mcherry*)[hu4881/s896] and scheme representation of CoW arteries in zebrafish brain at 32 hours post fertilization (hpf) (**A**), 54 hpf (**B**), 3 days post fertilization (dpf) (**C**), and 4 dpf (**D**). Green channel represents *flt4:yfp* fluorescence, red channel represents *kdrl:hras-mcherry* fluorescence, and merge panel combines both channels. Arrows point to the CoW arteries with *kdrl:hras-mcherry* signal. Scale bar = 50 μm. (**E**) Average intensity of *kdrl:hras-*

*Figure 1 continued on next page*

*Figure 1 continued*

*mcherry* in caudal division of internal carotid arteries (CaDI), basal communicating artery (BCA), and posterior communicating segments (PCS) at 32 hpf (n=6, 2 independent experiments), 54 hpf (n=12, 4 independent experiments), 3 dpf (n=15, 4 independent experiments), and 4 dpf (n=9, 2 independent experiments), two-way analysis of variance followed by Tukey's multiple comparisons, represented with mean ± SD, *p≤0.05, **p≤0.01. Abbreviations: hpf: hour post fertilization, dpf: day post fertilization, EC: endothelial cell, l-CaDI: left caudal division of internal carotid artery, r-CaDI: right caudal division of internal carotid artery, BCA: basal communicating artery, l-PCS: left posterior communicating segment, r-PCS: right posterior communicating segment.

The online version of this article includes the following source data for figure 1:

**Source data 1.** Source data for average intensity of kdrl:hras-mcherry in *Figure 1E*.

is consistent with CaDI being the inlet vessel for arterial blood flow within the entire CoW circulation during development. Next, we used computational fluid dynamics (CFD) to simulate the effect of flow on CoW, based on dextran-filled CoW geometry and RBC velocity (*ANSYS, 2014*; *Chen et al., 2019*; *Fernandes et al., 2022*, *Barak et al., 2021*). We found an overall increase in average WSS from 54 hpf to 4 dpf (*Figure 3A–C and H*, *Figure 3—source data 1*). Further, the simulated flow pattern and WSS in CoW showed that WSS in CaDI was significantly higher than BCA and PCS at 54 hpf, 3 dpf, and 4 dpf (*Figure 3A–C and I–K*, *Figure 3—source data 1*). Moreover, upon comparing WSS with the number of *acta2*+ VSMCs on CaDI during development, we observed a Pearson correlation coefficient of r=0.595, indicating a moderate to strong positive correlation between these two variables (*Figure 2—figure supplement 1C*). In summary, the analysis of blood flow suggests that CoW arteries bear different hemodynamic WSS, which correlates with the spatiotemporal pattern of VSMC differentiation on CoW arteries.

## Blood flow is required for CoW artery muscularization

Our data suggest that ECs under arterial flow might favor *acta2*+ VSMC differentiation from *pdgfrb*+ progenitors. To test this hypothesis, we set up an *in vitro* cell co-culture experiment where GFP-PDGFRB+ human brain vascular pericytes (GFP-HBVPs) were cultured in flow amenable outlet slides, then covered by a thin layer of collagen type 1. Confluent human endothelial cells (HUVECs) were seeded on top of that and then exposed to steady-state laminar or pulsatile flow conditions (*Figure 4A*; *Abello et al., 2022*). Differentiation of PDGFRB+ human brain pericytes into VSMCs was then quantified after 24 hours of flow treatment. We found that brain pericytes under pulsatile flow were larger and showed higher expression of VSMC differentiation markers including PDGFRB, ACTA2, and TRANSGELIN (*Robin et al., 2013*; *Figure 4B–D*, *Figure 4—source data 1*), suggesting that pulsatile flow favored PDGFRB+ progenitor differentiation into VSMC.

Previous research suggests that blood flow is dispensable for recruitment of *pdgfrb*+ progenitors on BCA and PCS (*Ando et al., 2016*). To corroborate this, we injected 0.35 ng of *tnnt2a* morpholino antisense oligonucleotide (MO) into Tg(*pdgfrb:egfp*, *kdrl:hras-mcherry*)[ncv22/s896] at one- to two-cell stage, to abrogate cardiac contractility and thus blood flow (*Ando et al., 2016*; *Chi et al., 2008*; *Sehnert et al., 2002*). Indeed, the number of *pdgfrb*+ cells on CaDI, BCA, and PCS at 3 dpf in *tnnt2a* morphants was comparable to uninjected sibling control, suggesting blood flow did not affect *pdgfrb*+ progenitor recruitment to CoW arteries (*Figure 4—figure supplement 1G, H*). To determine the effect of blood flow on VSMC differentiation, we injected *tnnt2a* MO into Tg(*acta2:mcherry*, *kdrl:gfp*)[ca8/zn1] (*Cross et al., 2003*; *Sehnert et al., 2002*; *Whitesell et al., 2014*). Notably, we found no *acta2*+ VSMCs on CoW arteries of *tnnt2a* morphants at either 3 dpf or 4 dpf (*Figure 4E, F, H, and I*, *Figure 4—source data 1*), suggesting that blood flow was required for *acta2*+ VSMC differentiation on CoW arteries.

To further explore the temporal requirement of blood flow, we treated embryos with 25 μM nifedipine, a drug shown to reduce heart rate in zebrafish embryos (*Figure 4—figure supplement 1A–C*; *Gierten et al., 2020*), from 54 hpf, right before the onset of *pdgfrb*+ progenitors' expression of *acta2* (*Figure 2A–C*). We found that at 3 dpf the number of *acta2*+ VSMCs on the CaDI of treated embryos is greatly reduced (*Figure 4E, G, and J*); *acta2*+ VSMCs on BCA and PCS at 4 dpf are also reduced significantly (*Figure 4E, G, and K*, *Figure 4—source data 1*), suggesting that blood flow reduction after 54 hpf impaired VSMC differentiation on CoW arteries.

To determine whether blood flow is required for short-term VSMC maintenance after differentiation on CoW arteries, we treated embryos with nifedipine from 4 dpf, after VSMC differentiation on

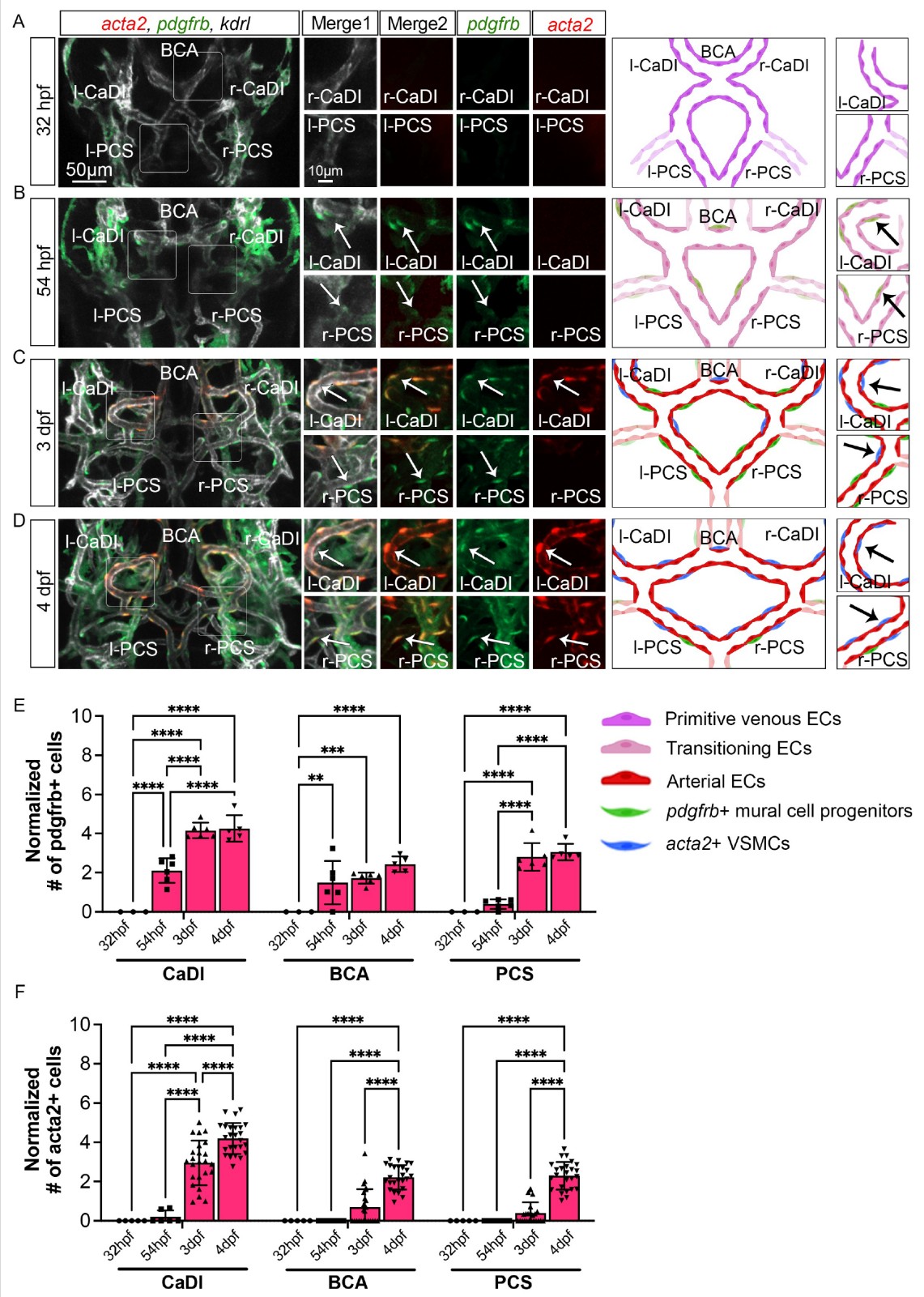

**Figure 2.** Vascular smooth muscle cell (VSMC) differentiation on circle of Willis (CoW) arteries. (**A–D**) Confocal live images of CoW arteries in Tg(*acta2:mcherry*, *kdrl:cerulean*)^ca8/sd24^ TgBAC(*pdgfrb:egfp*)^ncv22^ and scheme representation of vascular endothelium and mural cells on CoW arteries in zebrafish brain at 32 hours post fertilization (hpf) (**A**), 54 hpf (**B**), 3 days post fertilization (dpf) (**C**), and 4 dpf (**D**). White channel represents *kdrl:cerulean* fluorescence, red channel represents *acta2:mcherry*, green channel represents *pdgfrb:egfp,* merge 1 panel combines all three channels, merge 2

*Figure 2 continued on next page*

*Figure 2 continued*

combines *acta2:mcherry* in red and *pdgfrb:egfp* in green. Arrows point to the CoW arteries with *pdgfrb:egfp* and *acta2:mcherry* signal. Scale bar = 50 μm. (**E**) Number of *pdgfrb*+ vascular mural cell progenitors per 100 μm vessel length on caudal division of internal carotid arteries (CaDI), basal communicating artery (BCA), and posterior communicating segments (PCS) at 32 hpf (n=3, 1 independent experiment), 54 hpf (n=6, 1 independent experiment), 3 dpf (n=6, 1 independent experiment), and 4 dpf (n=5, 1 independent experiment), two-way analysis of variance (ANOVA) followed by Tukey's multiple comparisons, represented with mean ± SD, **p≤0.01, ***p≤0.001, ****p≤0.0001. (**F**) Number of *acta2*+ VSMCs per 100 μm vessel length on CaDI, BCA, and PCS at 32 hpf (n=5, 1 independent experiment), 54 hpf (n=6, 2 independent experiments), 3 dpf (n=24, 6 independent experiments), and 4 dpf (n=25, 6 independent experiments), two-way ANOVA followed by Tukey's multiple comparisons, represented with mean ± SD, ****p≤0.0001. Abbreviations: hpf: hour post fertilization, dpf: day post fertilization, EC: endothelial cell, VSMC: vascular smooth muscle cell, l-CaDI: left caudal division of internal carotid artery, r-CaDI: right caudal division of internal carotid artery, BCA: basal communicating artery, l-PCS: left posterior communicating segment, r-PCS: right posterior communicating segment.

The online version of this article includes the following source data and figure supplement(s) for figure 2:

**Source data 1.** Source data for number of *pdgfrb*+ vascular mural cell progenitors and *acta2*+ vascular smooth muscle cells (VSMCs) in *Figure 2E and F*.

**Figure supplement 1.** Vascular smooth muscle cell (VSMC) differentiation on circle of Willis (CoW) arteries.

CoW arteries, and imaged at 5 dpf. The number of *acta2*+ VSMCs on the CaDI and PCS in treated embryos was comparable to controls (*Figure 4—figure supplement 1D–F*), suggesting that blood flow was required for differentiation but not for short-term maintenance of VSMCs.

## Blood flow-regulated transcription factor *klf2a* modulates CoW artery muscularization

Our results suggest that CoW ECs might express a flow-dependent transcriptional program that regulates spatiotemporal dynamics of VSMC differentiation on CoW arteries. Previous studies showed that expression pattern of transcription factor *KLF2* in arterial ECs closely follows the pattern of flow-induced WSS (*Lee et al., 2006*; *Parmar et al., 2006*). Consistently, higher *KLF2* expression can be induced by unidirectional pulsatile flow (*Dekker et al., 2002*). Furthermore, *Klf2* is implicated in VSMC migration on arteries in mouse development: loss of *Klf2* leads to aorta VSMC deficiency (*Wu et al., 2008*). Interestingly, *klf2a* expression in primitive veins of zebrafish trunk prevents VSMC association (*Stratman et al., 2020*). Hence, we tested the hypothesis that *klf2a* might be spatiotemporally regulated in CoW ECs and modulate VSMC differentiation. We first imaged Tg(*klf2a:h2b-egfp*, *kdrl:hras-mcherry*)[ig11/s896], which labels nuclei with active *klf2a* signaling, and quantified the number of *klf2a*+ ECs in CoW arteries (*Chi et al., 2008*; *Steed et al., 2016*). To account for overall changes in EC number during development (*Ulrich et al., 2011*), and length of each vessel, we also imaged Tg(*fli1:nls-gfp*, *kdrl:hras-mcherry*)[y7/s896], which labels all EC nuclei (*Figure 5—figure supplement 1A–F*; *Roman et al., 2002*). From 32 hpf to 54 hpf, the number of *klf2a*+ ECs in CaDI was comparable to PCS (*Figure 5A–C*, *Figure 5—source data 1*), however, the number of *klf2a*+ ECs in CaDI was significantly higher than BCA and PCS from 3 dpf (*Figure 5A and D–E*, *Figure 5—source data 1*), when *acta2*+ VSMCs differentiated on CaDI but not BCA and PCS (*Figure 2C and F*, and *Figure 2—figure supplement 1A, B*). Pearson correlation between the number of *klf2a*+ ECs and the number of *acta2*+ VSMCs on CaDI, BCA, and PCS at 3 dpf showed strong positive correlation with r=0.974 (*Figure 5—figure supplement 1J*). These results suggest that the number of *klf2a*+ ECs in CoW arteries is spatially correlated with VSMC differentiation.

To further define the role of *klf2a* in VSMC differentiation on CoW arteries, we knocked down *klf2a* in Tg(*acta2:mcherry*, *kdrl:gfp*)[ca8/zn1] with 11 ng MO injection at one- to two-cell stage and imaged at 3–4 dpf (*Nicoli et al., 2010*; *Whitesell et al., 2014*; *Zhong et al., 2006*). We validated the effect of *klf2a* MO knockdown with Tg(*klf2a:h2b-egfp*, *kdrl:hras-mcherry*)[ig11/s896] (*Figure 5—figure supplement 1G–I*; *Chi et al., 2008*; *Steed et al., 2016*). Compared to uninjected control, *klf2a* morphants have significantly less *acta2*+ VSMCs on CaDI at 3 dpf but a normal number by 4 dpf (*Figure 6*, *Figure 6—source data 1*), suggesting that *klf2a* knockdown delayed VSMC differentiation on CaDI. Together these data suggest that endothelial *klf2a* promotes initial VSMC differentiation on anterior CoW arteries.

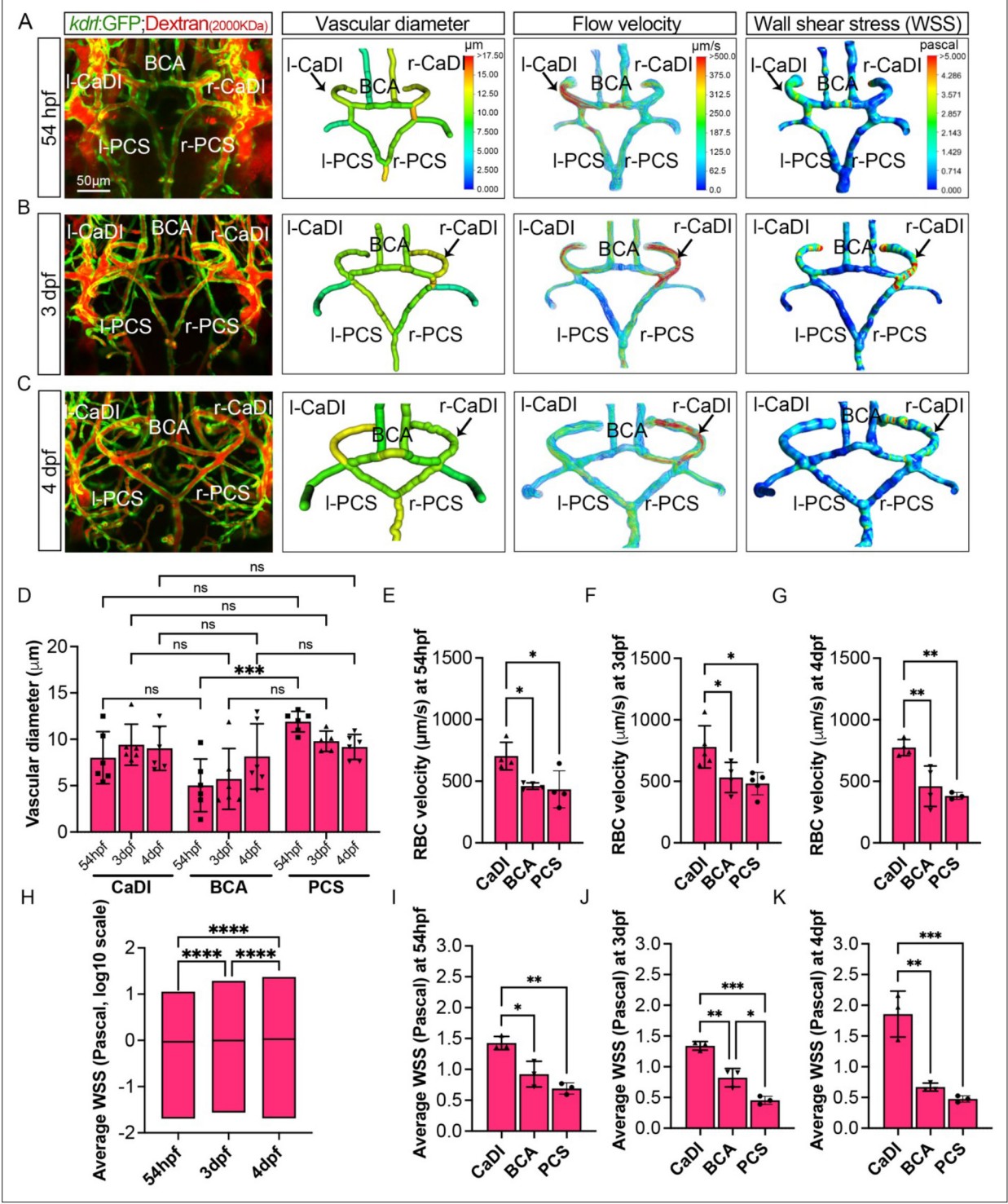

**Figure 3.** Computational fluid dynamic (CFD) simulation of circle of Willis (CoW) arteries. (**A–C**) Confocal live images of CoW arteries in Tg(*kdrl:gfp*)[zn1] injected with dextran and representation of vascular diameter, CFD simulated flow, and wall shear stress (WSS) at 54 hours post fertilization (hpf) (**A**), 3 days post fertilization (dpf) (**B**), and 4 dpf (**C**). Arrows point to CoW arteries with high flow velocity and WSS. Scale bar = 50 μm. (**D**) Average vascular diameter in caudal division of internal carotid arteries (CaDI), basal communicating artery (BCA), and posterior communicating segments (PCS) at 54 hpf (n=6, 1 independent experiment), 3 dpf (n=6, 1 independent experiment), and 4 dpf (n=6, 1 independent experiment), two-way analysis of variance (ANOVA) followed by Tukey's multiple comparisons, ***p≤0.001. (**E–G**) Red blood cell (RBC) velocity in CaDI, BCA, and PCS at 54 hpf (n=4, 1 independent experiment) (**E**), 3 dpf (n=5, 1 independent experiment) (**F**), and 4 dpf (n=4, 1 independent experiment) (**G**), ordinary one-way ANOVA with Tukey's multiple comparisons, represented with mean ± SD, *p≤0.05, **p≤0.01. (**H**) Average WSS throughout CoW arteries at 54 hpf (n=3 independent

*Figure 3 continued on next page*

*Figure 3 continued*

CFD simulation on CoW geometry), 3 dpf (n=3), and 4 dpf (n=3), ordinary one-way ANOVA with Tukey's multiple comparisons, ****p≤0.0001.
(**I–K**) Average WSS in CaDI, BCA, and PCS at 54 hpf (n=3) (**I**), 3 dpf (n=3) (**J**), and 4 dpf (n=3) (**K**), ordinary one-way ANOVA with Tukey's multiple comparisons, represented with mean ± SD, *p≤0.05, **p≤0.01, ***p≤0.001. Abbreviations: hpf: hour post fertilization, dpf: day post fertilization, RBC: red blood cell, WSS: wall shear stress, l-CaDI: left caudal division of internal carotid artery, r-CaDI: right caudal division of internal carotid artery, BCA: basal communicating artery, l-PCS: left posterior communicating segment, r-PCS: right posterior communicating segment.

The online version of this article includes the following source data for figure 3:

**Source data 1.** Source data for vascular diameter, red blood cell (RBC) velocity, and wall shear stress in *Figure 3D–K*.

## Discussion

In this study, we used confocal live imaging of fluorescence transgenic zebrafish embryos to investigate the spatiotemporal dynamics of VSMC differentiation on CoW, which comprises major arteries supplying blood to the vertebrate brain. Our observations revealed that CoW morphogenesis precedes arterial specification. Mural cell progenitors marked by *pdgfrb*+ initiate VSMC marker *acta2* expression subsequent to their recruitment to CoW arteries. Notably, VSMC differentiation occurs earlier on anterior CoW arteries compared to their posterior counterparts, owing to elevated WSS resulting from higher velocity of RBCs in the incoming blood flow. To investigate the regulatory role of blood flow on spatiotemporal dynamics of VSMC differentiation, we employed *in vitro* co-culture assay along with genetic manipulation and drug treatment. Our findings indicate that blood flow indeed governs timing and location of VSMC differentiation on CoW arteries. Moreover, we observed that flow-responsive transcription factor *klf2a* is activated in a gradient manner from anterior to posterior CoW arteries, preceding VSMC differentiation. Knockdown experiments targeting *klf2a* revealed a delay in VSMC differentiation specifically on anterior CoW arteries. Taken together, these findings highlight endothelial *klf2a* activation by blood flow as a mechanism that promotes VSMC differentiation on CoW arteries in the vertebrate brain (see *Figure 7* for a visual summary).

Previous research shows that in zebrafish trunk, as soon as recruited to dorsal aorta, VSMCs express *acta2* and *transgelin (tagln)*, suggesting simultaneous recruitment and differentiation from sclerotome progenitors (*Ando et al., 2016*; *Stratman et al., 2017*). On CoW arteries in zebrafish brain, however, VSMC differentiation occurs after *pdgfrb*+ progenitor recruitment and proceeds from anterior to posterior (*Figure 2B–F*). The observed spatiotemporal dynamics of CoW VSMC differentiation again highlights organotypic development of blood vessels.

Our data suggest *klf2a* mediates blood flow regulation of VSMC differentiation on brain arteries, and thus raise the question of how *klf2a* transduces endothelial signals to mural cell progenitors and VSMCs. Notch signaling appears a plausible downstream effector of *klf2a* activation, as previous research suggests Notch responds to flow in heart valve development (*Fontana et al., 2020*), possibly downstream of *klf2* (*Duchemin et al., 2019*). Wnt signaling is another possible downstream effector of *klf2a*, as previous research suggests endocardial *klf2* upregulates Wnt signaling in neighboring mesenchymal cells in heart valve development (*Goddard et al., 2017*). The roles of Notch and Wnt signaling in organotypic VSMC differentiation on brain arteries remain to be determined.

The expression of *klf2a* in CoW arterial ECs remained stable from 3 dpf to 4 dpf (*Figure 5*), which raises an interesting question on whether sustained *klf2a* expression supports further maturation of VSMCs, as *acta2*+ VSMCs on CaDI showed distinct morphology at 4 dpf compared with 3 dpf, when they started differentiation from *pdgfrb*+ mural cell progenitors (*Figure 2C, D* and *Figure 2—figure supplement 1A, B*). Previous research found that *acta2*+ VSMCs on BCA and PCS express pericyte enriched *abcc9* (ATP-binding cassette subfamily C member 9) at 4 dpf (*Ando et al., 2022*), when they started differentiation from *pdgfrb*+ mural cell progenitors and initiated *acta2* expression; *abcc9* expression is gradually lost from 5 dpf to 6 dpf (*Ando et al., 2022*). In addition, *acta2*+ VSMCs on CaDI, BCA, and PCS still retain expression of *pdgfrb* as late as 6 dpf (*Ando et al., 2021*). Thus, it is possible that stable expression of *klf2a* in CoW arterial ECs supports further VSMC maturation, as indicated by expression of *tagln* (*Colijn et al., 2023*). Another aspect of VSMC maturation is its function to confer vascular tone. A previous study found that VSMC covered vessels in zebrafish brain dilate as early as 4 dpf and constrict at 6 dpf (*Bahrami and Childs, 2020*). Future study may focus on the association between expression of different VSMC markers and VSMC functional maturation.

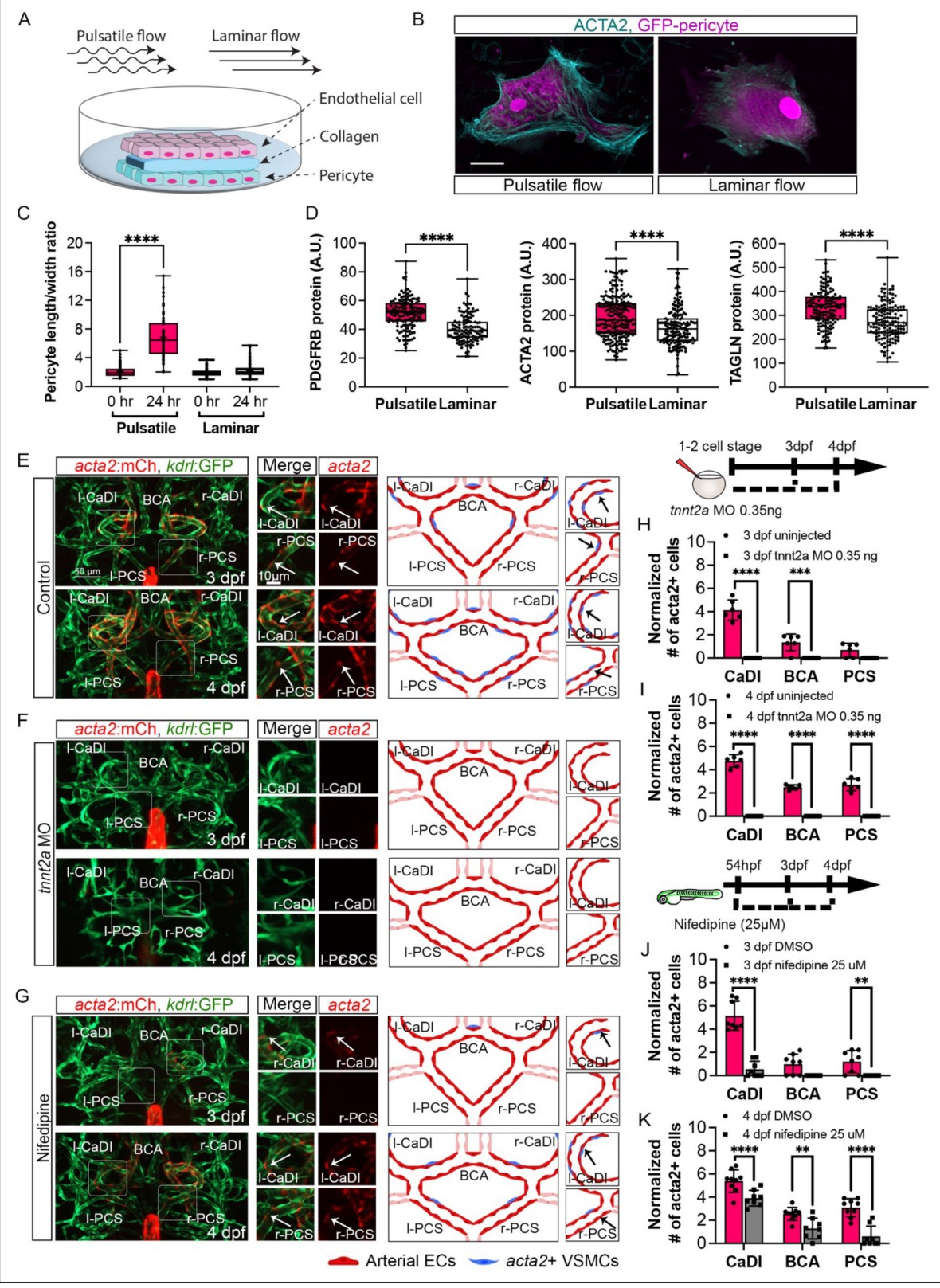

**Figure 4.** Blood flow is required for vascular smooth muscle cell (VSMC) differentiation on circle of Willis (CoW) arteries. (**A**) Scheme representation of *in vitro* cell co-culture experiment. (**B**) Representative immunofluorescence images of brain pericytes after exposure of pulsative flow and laminar flow. Cells were stained for ACTA2 (cyan) and cytosolic GFP label (magenta). Scale bar = 10 μm. (**C**) Morphological measurement of brain pericyte length/width ratio before and after exposure of pulsative flow and laminar flow (n=63 cells), two-tailed Mann–Whitney test, represented with mean

*Figure 4 continued on next page*

*Figure 4 continued*

± SD, ****p≤0.0001. (**D**) Protein level of PDGFRB (n=153 cells), ACTA2 (n=222 cells), and TAGLN (n=150 cells) in arbitrary unit (A.U.) after exposure of pulsative flow and laminar flow, two-tailed Mann–Whitney test, represented with mean ± SD, ****p≤0.0001. (**E–G**) Confocal live images of CoW arteries in Tg(acta2:mcherry; kdrl:gfp)ᶜᵃ⁸/ᶻⁿ¹ and scheme representation of vascular endothelium and VSMCs on CoW arteries at 3 days post fertilization (dpf) and 4 dpf in control embryos (**E**), embryos injected with 0.35 ng *tnnt2a* morpholino (MO) (**F**), and embryos treated with 25 µM nifedipine from 54 hpf (**G**). Red channel represents *acta2:mcherry*, green channel represents *kdrl:gfp*, and merge panel combines both channels. Arrows point to the CoW arteries with *acta2:mcherry* signal. Scale bar = 50 µm. (**H**) Number of *acta2*+ VSMCs per 100 µm vessel length on caudal division of internal carotid arteries (CaDI), basal communicating artery (BCA), and posterior communicating segments (PCS) at 3 dpf in uninjected control (n=6, 1 independent experiment) and embryos injected with 0.35 ng *tnnt2a* MO at one- to two-cell stage (n=6, 1 independent experiment), two-tailed Mann–Whitney test on each vessel's comparison, represented with mean ± SD, ***p≤0.001, ****p≤0.0001. (**I**) Number of *acta2*+ VSMCs per 100 µm vessel length on CaDI, BCA, and PCS at 4 dpf in uninjected control (n=6, 1 independent experiment) and embryos injected with 0.35 ng *tnnt2a* MO at one- to two-cell stage (n=6, 1 independent experiment), two-tailed Mann–Whitney test on each vessel's comparison, represented with mean ± SD, ****p≤0.0001. (**J**) Number of *acta2*+ VSMCs per 100 µm vessel length on CaDI, BCA, and PCS at 3 dpf in dimethyl sulfoxide (DMSO) control (n=8, 2 independent experiments) and embryos treated with 25 µM nifedipine from 54 hpf (n=8, 2 independent experiments), two-tailed Mann–Whitney test on each vessel's comparison, represented with mean ± SD, **p≤0.01, ****p≤0.0001. (**K**) Number of *acta2*+ VSMCs per 100 µm vessel length on CaDI, BCA, and PCS at 4 dpf in DMSO control (n=9, 2 independent experiments) and embryos treated with 25 µM nifedipine from 54 hpf (n=8, 2 independent experiments), two-tailed Mann–Whitney test on each vessel's comparison, represented with mean ± SD, **p≤0.01, ****p≤0.0001. Abbreviations: hpf: hour post fertilization, dpf: day post fertilization, EC: endothelial cell, VSMC: vascular smooth muscle cell, MO: morpholino, l-CaDI: left caudal division of internal carotid artery, r-CaDI: right caudal division of internal carotid artery, BCA: basal communicating artery, l-PCS: left posterior communicating segment, r-PCS: right posterior communicating segment.

The online version of this article includes the following source data and figure supplement(s) for figure 4:

**Source data 1.** Source data for brain pericyte length/width ratio, proteins level, and *acta2*+ vascular smooth muscle cells (VSMCs) number in *Figure 4C–D and H–K*.

**Figure supplement 1.** Blood flow is not required for vascular smooth muscle cell (VSMC) short-term maintenance or *pdgfrb*+ mural cell progenitor recruitment on circle of Willis (CoW) arteries.

**Figure supplement 1—source data 1.** Source data for heartbeat rate and number of *pdgfrb*+ vascular mural cell progenitors in *Figure 4—figure supplement 1C and H*.

---

Another question is how flow activates *klf2a* in brain arteries. In a hypoxia-induced pulmonary hypertension model, *Klf2* is activated by G-protein-coupled receptor-mediated Apelin signaling (*Chandra et al., 2011*). In heart valve development, endocardial *klf2a* is upregulated by membrane-bound mechanosensitive channels *trpp2* and *trpv4* (*Heckel et al., 2015*). The pathway through which flow activates *klf2a* in brain arterial ECs remains unknown. The CoW VSMC phenotype of *klf2a* morphants is similar to, but do not fully recapitulate, the phenotypes of *tnnt2a* morphants or nifedipine-treated embryos (*Figure 4E–K* and *Figure 6*). A proximal explanation is compensation by paralogous *klf2b* in zebrafish. Further characterization of CoW VSMC development in *klf2a* and *klf2b* genetic mutants (*Rasouli et al., 2018*; *Steed et al., 2016*) may help determine whether *klf2b* compensates *klf2a* in CoW VSMC differentiation.

In addition to WSS, transmural pressure and associated mechanical stretch is another mechanical input of vascular wall that may contribute to VSMC differentiation. VSMCs autonomously sense and adapt to mechanical stretch (*Haga et al., 2007*). Cyclic stretching activates arterial VSMC production of certain components of extracellular matrix (ECM) (*Leung et al., 1976*). In turn, specific interactions between ECM and integrins enable VSMCs to sense mechanical stretch (*Goldschmidt et al., 2001*; *Wilson et al., 1995*). Connection to ECM is essential for VSMC force generation (*Milewicz et al., 2017*). VSMCs express stretch activated *Trpp2*, which enables myogenic response to autoregulate resting arterial diameter (*Sharif-Naeini et al., 2009*). Interestingly, *ex vivo* stretching promotes expression of contractile proteins, such as *Acta2* and *Tagln* in VSMCs on murine portal veins (*Albinsson et al., 2004*; *Turczyńska et al., 2013*). In addition, excessive mechanical stretch promotes VSMC dedifferentiation and inflammation (*Cao et al., 2017*; *Wang et al., 2018*). Further investigation is needed to determine the potential role of increasing transmural pressure and associated mechanical stretch during development in VSMC differentiation.

There are a few limitations to our current study. Genetic manipulation and drug treatment that reduce heart rate would also reduce nutrients carried by blood flow, and the effects of nutrient and flow reduction could not be uncoupled in live zebrafish embryos. In addition, these methods are only capable of qualitative reduction of flow but not specific dampening of pulsations. *In vitro* three-dimensional (3D) vascular culture models, which combine ECs and mural cells (*Mirabella et al., 2017*;

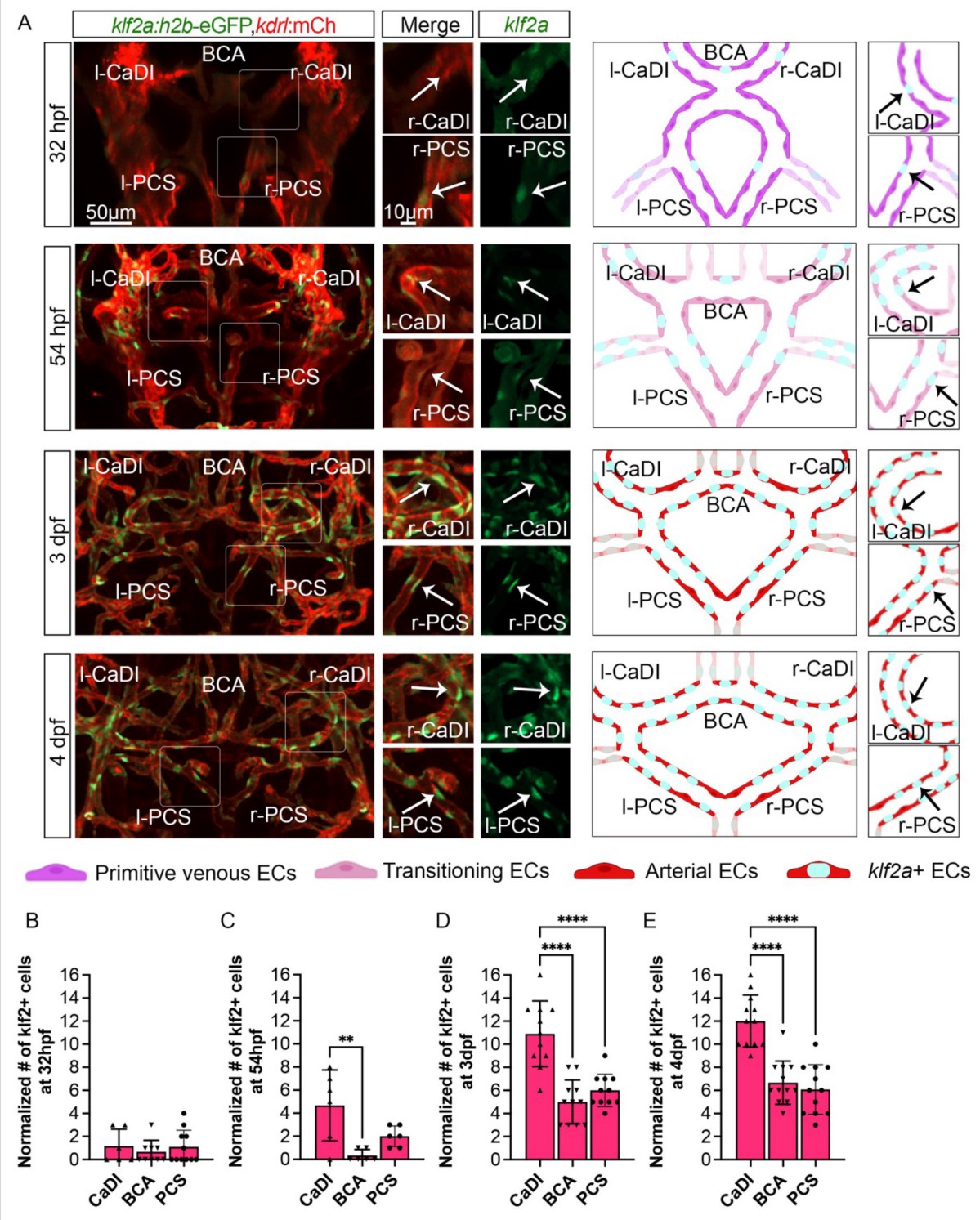

**Figure 5.** Blood flow-regulated transcription factor *klf2a* is expressed in circle of Willis (CoW) arteries. (**A**) Confocal live images of CoW arteries in Tg(*klf2a:h2b-gfp*, *kdrl:ras-mcherry*)[ig11/s896] and scheme representation of endothelial cells (ECs) in CoW arteries at 32 hours post fertilization (hpf), 54 hpf, 3 days post fertilization (dpf), and 4 dpf. Green channel represents *klf2a:h2b-gfp*, red channel represents *kdrl:ras-mcherry*, and merge panel combines both channels. Arrows point to the CoW arteries with *klf2a:h2b-gfp* signal. Scale bar = 50 µm. (**B–E**) Number of *klf2a*+ ECs per 100 µm vessel length on caudal division of internal carotid arteries (CaDI), basal communicating artery (BCA), and posterior communicating segments (PCS) at 32 hpf (n=6, 3 independent experiments) (**B**), 54 hpf (n=6, 2 independent experiments) (**C**), 3 dpf (n=11, 3 independent experiments) (**D**), and 4 dpf (n=12, 3 independent experiments) (**E**), ordinary one-way analyses of variance with Tukey's multiple comparisons, represented with mean ± SD, **p≤0.01,

*Figure 5 continued on next page*

*Figure 5 continued*

****p≤0.0001. Abbreviations: hpf: hour post fertilization, dpf: day post fertilization, EC: endothelial cell, l-CaDI: left caudal division of internal carotid artery, r-CaDI: right caudal division of internal carotid artery, BCA: basal communicating artery, l-PCS: left posterior communicating segment, r-PCS: right posterior communicating segment.

The online version of this article includes the following source data and figure supplement(s) for figure 5:

**Source data 1.** Source data for number of *klf2a*+ endothelial cells (ECs) in *Figure 5B–E*.

**Figure supplement 1.** Number of endothelial cells (ECs) in circle of Willis (CoW) arteries does not increase significantly during *klf2a* activation.

**Figure supplement 1—source data 1.** Source data for number of endothelial cells (ECs) and average intensity of *klf2a:h2b-gfp* in *Figure 5—figure supplement 1C–F and H–I*.

*Vila Cuenca et al., 2021*), could be further optimized to simulate complex geometry of brain arteries. Combining these models with microfluidics, which allows precise calibration of nutrient composition in culture media, flow velocity, and pulse, would enable more thorough analysis of endothelial mechanotransduction and its contribution to VSMC differentiation (*Abello et al., 2022*; *Gray and Stroka, 2017*; *Griffith et al., 2020*).

We used nifedipine to temporally reduce heart rate and blood flow. Nifedipine is a blocker of L-type voltage-dependent calcium channels (VDCCs) (*Quevedo et al., 1998*). A previous study shows that ML218, a T-type VDCC-selective inhibitor, tends to increase VSMC differentiation (*Ando et al., 2022*). In addition to the difference in drug targets, we also noted that in the previous study, the increase in VSMC differentiation only occur on anterior metencephalic central arteries (AMCtAs) that are more than 40 μm away from the BCA; these AMCtAs are much smaller than CoW arteries and have different geometry (*Ando et al., 2022*). Although the most obvious effect of nifedipine in zebrafish embryos is heart rate and blood flow reduction (*Gierten et al., 2020*), it is possible that nifedipine affects VSMC through VDCCs. Further investigation is needed to uncouple the roles of different VDCCs in VSMC differentiation and their association with hemodynamics.

We used Tg(*flt4:yfp*, *kdrl:ras-mcherry*)[hu4881/s896] to show CoW arterial specification from primitive venous ECs (*Chi et al., 2008*; *Hogan et al., 2009*). *kdrl* is not the best arterial marker as its expression is only enriched but not restricted to arteries. While the use of the two fluorescence transgenic lines helped us establish the temporal sequence of CoW morphogenesis, arterial specification and VSMC differentiation, analysis of additional arterial and venous markers is needed to fully characterize arterial specification in vertebrate brain vascular development.

In summary, our work identifies flow-induced endothelial *klf2a* activation as a mechanism that regulates spatiotemporal dynamics of VSMC differentiation on CoW arteries in vertebrate brain. Our data may help advance approaches to regenerate dedifferentiated VSMCs in cerebrovascular disease. It would be important to further investigate mechanisms upstream and downstream of *klf2a* activation in brain arterial ECs and additional mechanotransduction pathways that may contribute to VSMC differentiation on brain arteries.

## Materials and methods
### Zebrafish husbandry and transgenic lines

Zebrafish were raised and maintained at 28.5°C using standard methods. Protocols are approved by the Yale Institutional Animal Care and Use Committee (2020-11473). TgBAC(*pdgfrb:egfp*)[ncv22] was generated by *Ando et al., 2016*. They reported that *pdgfrb:egfp*+ cells express another pericyte enriched marker *abcc9* (ATP-binding cassette subfamily C member 9) (*Ando et al., 2019*, *Ando et al., 2022*). With this transgenic line, they also observed that *pdgfrb* mutant lacks brain pericytes and exhibits loss of VSMC coverage (*Ando et al., 2021*). Thus, we used TgBAC(*pdgfrb:egfp*)[ncv22] to study mural cell coverage of vascular endothelium in zebrafish CoW arteries. All transgenic lines in *Table 1* were established previously.

### Confocal fluorescence microscopy

Zebrafish embryos were raised in 0.003% 1-phenyl-2-thiourea (PTU, phenylthiocarbamide, or *n*-phenylthiourea, Sigma P7629) from gastrulation stage to prevent pigmentation. Embryos imaged live by confocal fluorescence microscopy were anesthetized in 0.1% tricaine methanesulfonate (TMS,

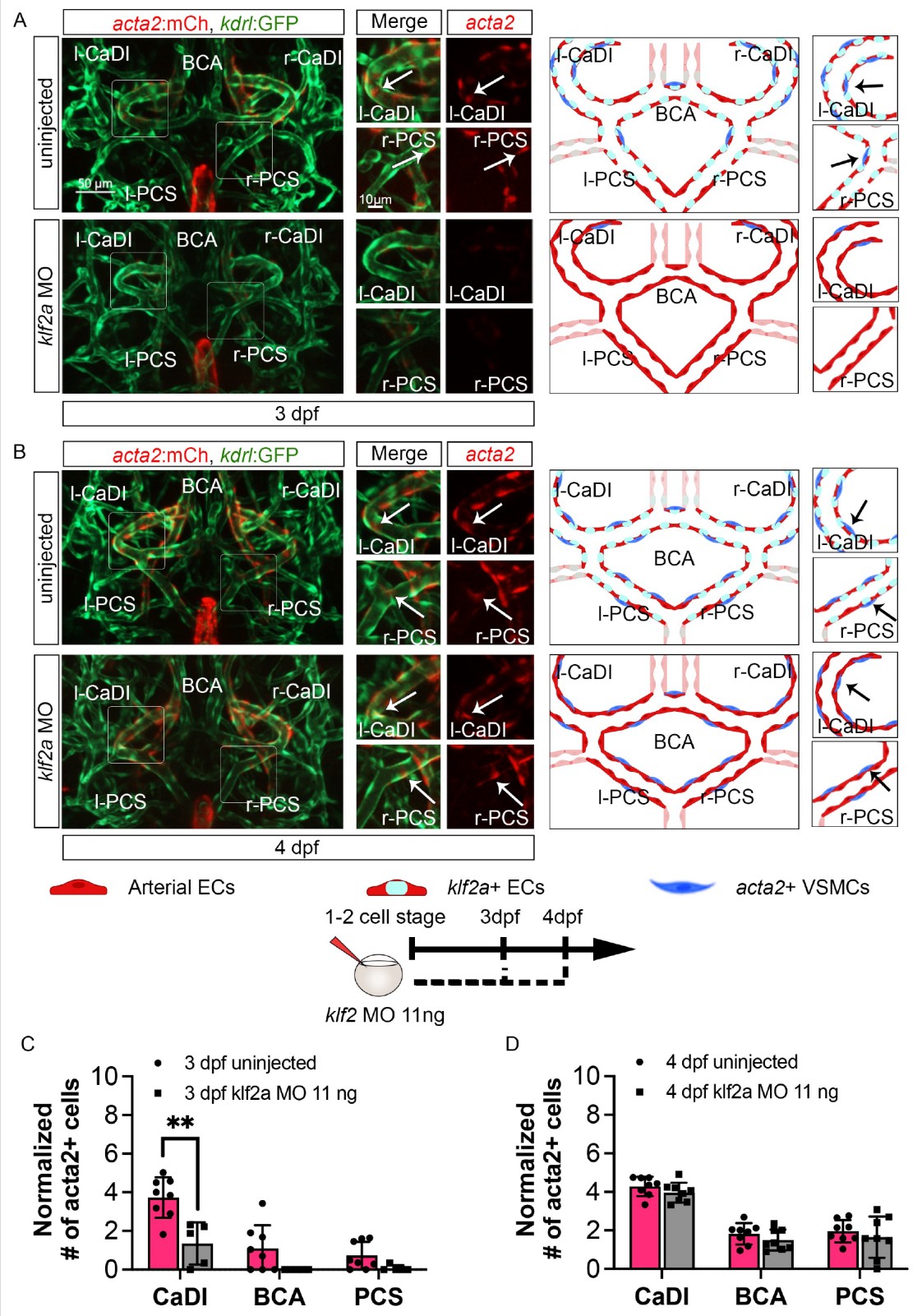

**Figure 6.** *klf2a* promotes vascular smooth muscle cell (VSMC) differentiation on anterior circle of Willis (CoW) arteries. (**A–B**) Confocal live images of CoW arteries in Tg(*acta2:mcherry; kdrl:gfp*)[ca8/zn1] and scheme representation of vascular endothelium and VSMCs on CoW arteries at 3 days post fertilization (dpf) (**A**) and 4 dpf (**B**) in uninjected control embryos and embryos injected with 11 ng *klf2a* morpholino (MO). Red channel represents *acta2:mcherry*, green channel represents *kdrl:gfp*, and merge panel combines both channels. Arrows point to the CoW arteries with *acta2:mcherry*

*Figure 6 continued on next page*

*Figure 6 continued*

signal. Scale bar = 50 μm. (**C**) Number of *acta2*+ VSMCs per 100 μm vessel length on caudal division of internal carotid arteries (CaDI), basal communicating artery (BCA), and posterior communicating segments (PCS) at 3 dpf in uninjected control (n=8, 2 independent experiments) and embryos injected with 11 ng *klf2a* MO at one- to two-cell stage (n=5, 2 independent experiments), two-tailed Mann–Whitney test on each vessel's comparison, represented with mean ± SD, **p≤0.01. (**D**) Number of *acta2*+ VSMCs per 100 μm vessel length on CaDI, BCA, and PCS at 4 dpf in uninjected control (n=8, 3 independent experiments) and embryos injected with 11 ng *klf2a* MO at one- to two-cell stage (n=8, 3 independent experiments), two-tailed Mann–Whitney test on each vessel's comparison, represented with mean ± SD. Abbreviations: hpf: hour post fertilization, dpf: day post fertilization, EC: endothelial cell, VSMC: vascular smooth muscle cell, MO: morpholino, l-CaDI: left caudal division of internal carotid artery, r-CaDI: right caudal division of internal carotid artery, BCA: basal communicating artery, l-PCS: left posterior communicating segment, r-PCS: right posterior communicating segment.

The online version of this article includes the following source data for figure 6:

**Source data 1.** Source data for *acta2*+ vascular smooth muscle cells (VSMCs) number in *Figure 6C and D*.

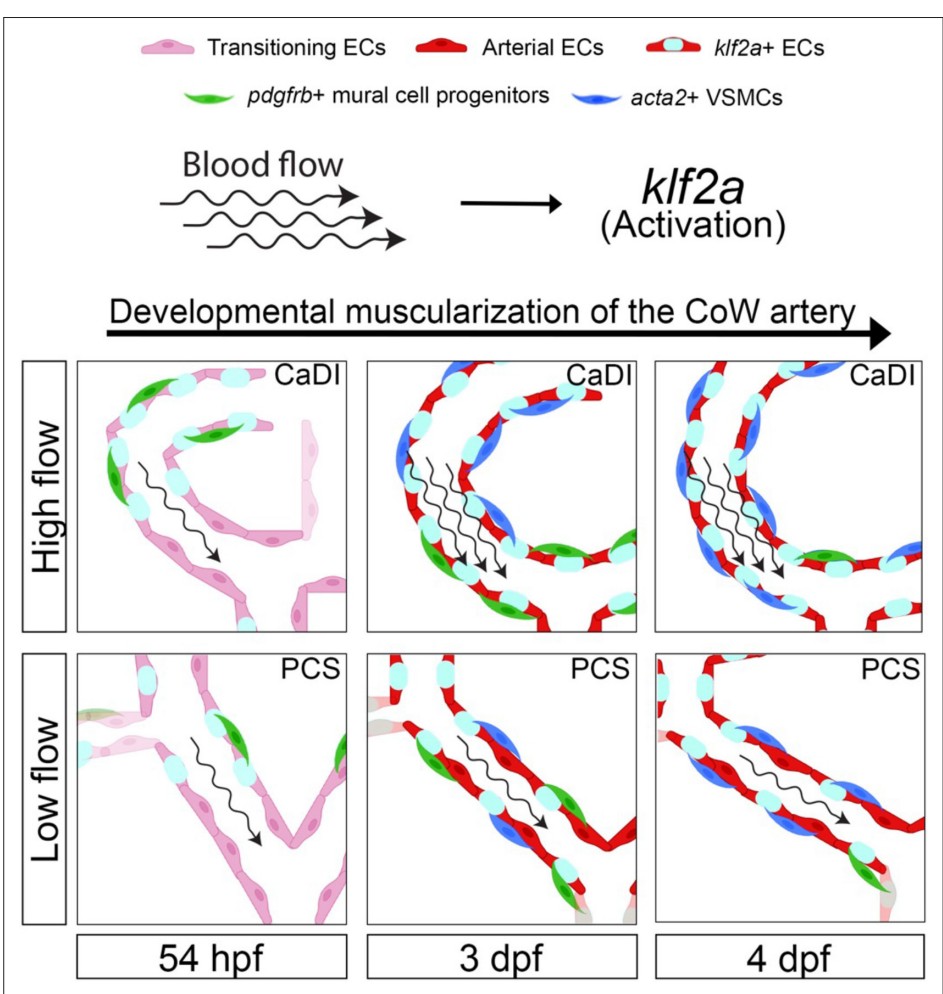

**Figure 7.** Schematic model of the developmental muscularization of circle of Willis (CoW) arteries. The model shows how blood flow generates higher hemodynamics in anterior CoW arteries like caudal division of internal carotid artery (CaDI) and activates endothelial *klf2a* signaling. Other posterior CoW arteries with straight shape like posterior communicating segment (PCS) bear less hemodynamic force and show moderate *klf2a* activation and later VSMC differentiation. Abbreviations: hpf: hour post fertilization, dpf: day post fertilization, EC: endothelial cell, VSMC: vascular smooth muscle cell, CaDI: caudal division of internal carotid artery, PCS: posterior communicating segment.

**Table 1.** List of zebrafish fluorescent transgenic lines used in the study.

| Transgenic line | ID | Lab of origin | Reference |
|---|---|---|---|
| Tg(flt4:yfp)[hu4881] | ZDB-ALT-100208-1 | S Schulte-Merker | *Hogan et al., 2009* |
| Tg(kdrl:hras-mcherry)[s896] [Tg(kdrl:ras-mcherry)[s896]] | ZDB-ALT-081212-4 | DYR Stainier | *Chi et al., 2008* |
| Tg(acta2:mcherry)[ca8] | ZDB-ALT-120508-2 | SJ Childs | *Whitesell et al., 2014* |
| TgBAC(pdgfrb:egfp)[ncv22] | ZDB-ALT-160609-1 | N Mochizuki | *Ando et al., 2016* |
| Tg(kdrl:cerulean)[sd24] | ZDB-ALT-131024-2 | D Traver | *Page et al., 2013* |
| Tg(kdrl:grcfp)[zn1] [Tg(kdrl:gfp)[zn1]] | ZDB-ALT-051114-10 | Zygogen Research Department | *Cross et al., 2003* |
| Tg(klf2a:h2b-egfp)[ig11] | ZDB-ALT-161017-10 | J Vermot | *Steed et al., 2016* |
| Tg(fli1:negfp)[y7] [Tg(fli1:nls-gfp)[y7]] | ZDB-ALT-060821-4 | BM Weinstein | *Roman et al., 2002* |
| Tg(gata1:dsRed)[sd2] | ZDB-ALT-051223-6 | LI Zon | *Traver et al., 2003* |

.

MS-222, or Syncaine, Western Chemical, NC0872873) and mounted in 1% low melt agarose within glass-bottom microwell dishes. Fluorescence images were captured with an upright Zeiss LSM 980 confocal microscope using a 20× objective.

## Image analysis

Confocal fluorescence images were analyzed with Imaris microscopy image analysis software (Bitplane, Oxford Instruments). Average fluorescence intensity of mCherry in Tg(flt4:yfp, kdrl:hras-mcherry)[hu4881/s896] was estimated by generating volume objects covering the artery of interest with Surface module. Vessel lengths of the arteries were manually traced with Filament module. Numbers of *pdgfrb*+ mural cell progenitors, *acta2*+ VSMCs, and *klf2a*+ and *fli1*+ endothelial nuclei were counted with Spots module.

## Morpholino injections

Morpholino antisense oligonucleotides were synthesized by Gene Tools. Morpholinos in *Table 2* were validated previously. Optimized dose injected into each embryo are listed. Uninjected siblings were used as controls.

## Nifedipine treatment

Nifedipine was dissolved into 20 mM in dimethyl sulfoxide (DMSO). The stock solution was diluted into 25 µM in egg water with 0.003% PTU to treat zebrafish embryos from 54 hpf to 3 dpf or 4 dpf. The stock solution was diluted into 20 µM in egg water with 0.003% PTU to treat zebrafish embryos from 4 dpf to 5 dpf. The same volume of DMSO was added into egg water with 0.003% PTU for sibling control embryos.

## Microangiography

Embryos were anesthetized in 0.1% TMS, placed on an agarose mold, and injected pericardially with 4 nL tetramethylrhodamine dextran (2,000,000 molecular weight, Thermo Fisher) at a concentration

**Table 2.** List of morpholino antisense oligonucleotides used in the study.

| Morpholino | Sequence | ID | Reference | Dose (ng/embryo) |
|---|---|---|---|---|
| MO1-tnnt2a | 5' - CATGTTTGCTCTGATCTGACACGCA -3' | ZDB-MRPHLNO-060317-4 | *Sehnert et al., 2002* | 0.35 |
| MO1-klf2a | 5' - GGACCTGTCCAGTTCATCCTTCCAC -3' | ZDB-MRPHLNO-100610-8 | *Nicoli et al., 2010* | 11 |

of 10 mg/mL. Subsequently, the vasculature of embryos was checked for dextran fluorescence signal under a stereomicroscope (Olympus, MVX10). Embryos with dextran fluorescence were mounted in 1% low melt agarose within glass-bottom microwell dishes and imaged with confocal microscope.

## RBC velocity

RBC velocity was measured with axial line scanning to analyze zebrafish in Tg(*kdrl:gfp;gata1:DsRed*)[zn1/sd2] background, in which blood vessels and RBCs are labeled with green and red fluorescent protein, respectively (*Cross et al., 2003*; *Traver et al., 2003*). As described previously (*Barak et al., 2021*), this axial line scan was conducted in a Zeiss LSM980 confocal microscope. A line was drawn inside the lumen of CoW vessels and parallel to the longitudinal axis of the vessel. The average length of the line was 21.21 µm. The scan speed was chosen as maximum per second for 20,000 cycles (total scan duration was 5.86 s; line scan time is 292.80 µs). A kymograph was exported in TIFF format after every measurement by ZEN 3.5 and analyzed RBC recognition and segmentation by using Fiji (ImageJ). First, the RBC kymography image was processed using 'noise despeckle' and 'enhance contrast' steps. The brightness and contrast were appropriately adjusted if needed. Next, the RBCs were segmented, thresholding was adjusted, and the slope of the RBCs was calculated automatically using the 'Analyze particle' option. Finally, the velocity of each RBC was calculated using the following equation: velocity = (1/tan(a)) * (length of the y-axis of the kymograph/duration of the total scan). We averaged systolic peak velocities and end diastolic velocities to measure the mean RBC velocity in CoW arteries.

## CFD simulation

CFD simulation was performed according to CoW geometry at 54 hpf, 3 dpf, and 4 dpf as previously described with modification (*ANSYS, 2014*; *Barak et al., 2021*). First, 3D geometry models of CoW arteries were reconstructed from confocal z-stack microangiography images of dextran-injected Tg(*kdrl:gfp*) embryos with Filament module in Imaris (9.9.0, Bitplane, Oxford Instruments). These CoW 3D geometry models were pre-processed in ANSYS SpaceClaim 2022 R1 software. For example, the computational mesh on the CoW vascular wall was created and the boundary conditions were set up for the simulation. Meshing orthogonal quality was calculated, and objects less than 0.01 were excluded. Inlets and outlets were determined in the 3D model for flow simulation. Next, the pre-processed models were imported in ANSYS Fluent 2022 R1 software. The non-Newtonian behavior of blood flow was modeled using the Carreau model provided by ANSYS. Blood density was considered as 1060 kg/m$^3$. The computational simulation used average RBC velocity in CoW arteries at 54 hpf (532.72 µm/s), 3 dpf (602.34 µm/s), and 4 dpf (552.83 µm/s), respectively. After the CoW geometry and numerical solution of blood flow were ready, the flow simulation was implemented with 200 iterations to identify the magnitude and distribution of WSS in the CoW.

## Cell culture

Pooled primary HUVECs (PCS-100-010, ATCC) were seeded at an initial concentration of 5000 cells/cm$^2$, in 1× M119 media (110343-023, Gibco) supplemented with 16% FBS (Gibco), 84 µg/mL of heparin sodium salt (H3393, Sigma-Aldrich), 25 µg/mL of EC growth supplement (02-102, EMD Millipore Corp.) and 1× Antibiotic-antimycotic solution (15240-062, Gibco). Cell cultures were maintained at 37°C, 5% CO$_2$, and 95% humidity, until the cells reached 80% confluence.

To facilitate cell visualization in cell co-cultures, HBVPs (1200, ScienCell) were transfected using lentiviral particles (LVP310, GenTarget, Inc) to induce GFP expression under EF1a promotor. Cell cultures were initiated by seeding 5000 cells/cm$^2$ in 175 cm$^2$ plastic flasks precoated with gelatin and 1× DMEM (11995-065, Gibco) supplemented with 10% FBS (Gibco), and 1× Antibiotic-antimycotic solution under the same cell culture conditions described above. When cell cultures were at 80% confluence, 200 µL containing 2×10$^6$ GFP-lentiviral particles were added to each 175 flasks. After 72 hours, fresh cell culture media supplemented with 10 µg/mL of Blasticidin (15205, Sigma-Aldrich) were added to each flask to select the positive transfected cells.

## Flow assays and immunostaining

HBVP cells were harvested and seeded at a concentration of 1.3×10$^5$ cells in 0.4 optical plastic flow microslides (80176, Ibidi) precoated with 1 mg/mL gelatin and incubated for 24 hours under standard culture conditions. After the initial incubation, 100 µg/mL of collagen I (354249, Corning) diluted in

DMEM cell culture media was added to the slides to create a thin layer on top of the HBVP cells. After 2 hours, the media was removed and $2.5 \times 10^5$ HUVECs were seeded on top of the collagen I layer and incubated for additional 24 hours. After cell co-cultures were established, the slides were exposed to laminar (15 dyn/cm$^2$) or pulsatile (12–15 dyn/cm$^2$) flow for 24 hours, implementing a peristaltic pump adapted to produce different types of flow (*Abello et al., 2022*). After 24 hours, cultures were rinsed with 1× PBS and fixed for 30 min in 4% paraformaldehyde at room temperature. Cell cultures were immunostained with alpha-smooth muscle actin D4K9N XP rabbit monoclonal antibody (19245, Cell Signaling), followed by Alexa Fluor 633 goat anti-rabbit IgG (A21071, Invitrogen). Confocal images were obtained using a 40× objective with a W1 Spinning Disk confocal microscope, a Fusion camera, and the Nikon Eclipse Ti2-E base. Fiji image processing software was used for image analysis and fluorescence intensity quantification.

## Statistics

All statistical analyses were performed with GraphPad Prism (version 10.0.3). Sample size and statistical test are specified in each legend. All quantifications are represented with mean ± SD. ns, not significant $p > 0.05$, $*p \leq 0.05$, $**p \leq 0.01$, $***p \leq 0.001$, $****p \leq 0.0001$. Mann–Whitney test was used to compare two groups to test mean differences (protein level, morpholino, and nifedipine treatment). Two-way analysis of variance (ANOVA) followed by Tukey's multiple comparisons was used to test mean differences compared more than two groups under different conditions (average fluorescence intensity, cell number, vascular diameter). Ordinary one-way ANOVA with Tukey's multiple comparisons was used to test mean differences among three groups (RBC velocity, WSS, and cell number). Pearson correlation analysis was used to measure the strength and direction of a linear relationship between two variables with a covariance (WSS vs *acta2*+ EC, *acta2*+ EC vs *klf2*+ EC).

## Acknowledgements

We thank the labs of S Schulte-Merker, DYR Stainier, SJ Childs, N Mochizuki, D Traver, J Vermot, BM Weinstein, and LI Zon for sharing zebrafish transgenic lines. We thank N Semanchik for all the assistance with zebrafish adult colonies and husbandry. Experiments in the manuscript were supported by R01NS109160 and R01DK118728 awarded to SN and AHA23POST1025829 post-doctoral fellowship awarded to IFX. The paper is based on a dissertation submitted by SC to fulfill in part the requirements for the degree of Doctor of Philosophy, Yale University. SC was supported by a Gruber Science Fellowship from Yale Graduate School of Arts and Sciences.

## Additional information

### Funding

| Funder | Grant reference number | Author |
| --- | --- | --- |
| National Institutes of Health | R01NS109160 | Stefania Nicoli |
| National Institutes of Health | R01DK118728 | Stefania Nicoli |
| American Heart Association | AHA23POST1025829 | Ivan Fan Xia |

The funders had no role in study design, data collection and interpretation, or the decision to submit the work for publication.

### Author contributions

Siyuan Cheng, Ivan Fan Xia, Conceptualization, Data curation, Software, Formal analysis, Validation, Investigation, Visualization, Methodology, Writing - original draft, Writing - review and editing; Renate Wanner, Methodology; Javier Abello, Data curation, Software, Formal analysis, Validation, Investigation, Visualization, Methodology; Amber N Stratman, Data curation, Software, Formal analysis, Supervision, Validation, Investigation, Visualization, Methodology, Writing - original draft, Writing - review

and editing; Stefania Nicoli, Conceptualization, Resources, Data curation, Formal analysis, Supervision, Funding acquisition, Investigation, Visualization, Methodology, Writing - original draft, Project administration, Writing - review and editing

### Author ORCIDs
Siyuan Cheng (iD) http://orcid.org/0000-0001-5013-7767
Ivan Fan Xia (iD) http://orcid.org/0000-0001-6607-5003
Amber N Stratman (iD) http://orcid.org/0000-0002-8111-4186
Stefania Nicoli (iD) http://orcid.org/0000-0001-7055-340X

### Ethics

This study was performed in strict accordance with the recommendations in the Guide for the Care and Use of Laboratory Animals of the National Institutes of Health. Protocols are approved by the Yale Institutional Animal Care and Use Committee (IACUC #2020-11473).

Reviewer #2 (Public review): https://doi.org/10.7554/eLife.94094.3.sa1
Reviewer #3 (Public review): https://doi.org/10.7554/eLife.94094.3.sa2
Author response https://doi.org/10.7554/eLife.94094.3.sa3

---

## Additional files

### Supplementary files
• MDAR checklist

### Data availability

All data generated or analysed during this study are included in the manuscript and supporting files; source data files have been provided for the figures.

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
