## [Editor Report · eLife assessment]

This study provides the first analysis of vascular stabilization on the critical and evolutionarily conserved structure around the Circle of Willis in the brain, strengthened by using parallel in vivo and in vitro experimental approaches. The evidence supporting the claims is **solid** and the work will be **valuable** for scientists studying developmental and disease-related vascular stabilization.

---

## [Referee Report · Reviewer #2 (Public review)]

Summary:

Cheng et al. explore the development of the arteries that form the circle of Willis and investigate how blood flow pulsatility influences vascular smooth muscle cell (VSMC) differentiation. Using live confocal imaging of the developing zebrafish, the authors show that endothelial cells in circle of Willis arteries transition from venous to arterial identity between 54 hours post-fertilization (hpf) and 3 days post-fertilization (dpf), and that this coincides with pdgfrb+ mural cell progenitor differentiation into acta2+ arterial VSMCs. They find that the anterior portions of the circle of Willis, including the internal carotid arteries (CaDI), establish acta2 expression earlier than posterior aspects, likely due to faster flow rate and increased pulsatility through the CaDI. Then, using computational fluid dynamics, an in vitro co-culture assay, and genetic and drug manipulations of blood flow, the authors provide evidence that pdgfrb+ differentiation is dependent upon pulsatile blood flow and klf2a activation. The results add to our understanding of vascular development and suggest that deficits in pulsatile flow could be potential drivers of arteriopathies.

Strengths:

(1) Longitudinal confocal imaging of live developing zebrafish makes the timeline of arterial development in the circle of Willis easy to understand. This is a strong approach to studying how vascular networks are altered with genetic and pharmacological manipulations.

(2) Rigorous use of multiple techniques to test the hypothesis that pulsatile blood flow is required for smooth muscle cell differentiation. The microangiography experiment, in vitro co-culture assay, and genetic and drug manipulations of heart rate at various developmental timepoints yield outcomes that are consistent with the hypothesis.

Weaknesses:

(1) The authors should provide more information on how blood flow velocity and wall shear stress are calculated from circle of Willis vascular structure. It is presumed that these values are dependent upon the 3-D morphology of the vessel network, as labeled by intravenous dextran dye, but this is not clear. Small local differences in vessel diameter and shape will influence blood flow velocity, but these morphological changes are not clearly articulated. Further, it is unclear how flow input levels to the CaDI and basilar arteries are decided across time-points. In general, descriptions of the blood flow modeling are very sparse.

(2) Is it possible to measure the blood flow speed empirically with line-scanning or high-speed tracking of labeled blood cells? This would provide some validation of the modeling results.

(3) Does the cardiac injection of dextran itself affect the diameter or flow of the arteries, given the invasiveness of the procedure? This could be examined in fish with a transgenic endothelial label and with vs. without dextran.

(4) The data from the microangiography experiment in Figure 3 does not fully support the stated results. The authors report that the CaDI had the highest blood flow speed starting from 54 hfp, but it does not appear to be higher than the other arteries at this time point. Additionally, there is not sufficient evidence that wall shear stress coincides with smooth muscle cell differentiation in the CaDI. Wall shear stress appears to be similar between 54 hpf and 3 dpf in the CaDI, only increasing between 3 dpf and 4 dpf, while differentiation is shown to begin at 3 dpf.

(5) The genetic and drug manipulations of heart rate are important experiments, but more detail is required to understand the effects of the manipulations. At least, a discussion on the limitations of these manipulations is needed. For example, how does one separate the pulsatile versus nutritive effects of blood flow/heart rate reduction? It is possible that off-target or indirect effects of Nifedipine decrease smooth muscle cell proliferation, or that altered cardiac contractility fundamentally alters many aspects of vascular development other than blood flow. Nifedipine is also likely to act upon VSMC calcium handling in the circle of Willis, which may in turn affect cell maturation.

(6) It is unclear if acta2 expression is conferring vascular tone, as would be expected if the cells are behaving as mature VSMCs. Does arterial diameter decrease with an increase in acta2 expression? Are acta2 positive mural cells associated with more dynamic changes in arteriole diameter under basal or stimulated conditions?

---

## [Referee Report · Reviewer #3 (Public review)]

Summary:

Cheng et al. studied if and how blood flow regulates differentiation of vascular smooth muscle cells (VSMC) in the Circle of Willis (CW) in zebrafish embryos. They show that CW vessels gradually acquire arterial identity. VSMCs also undergo gradual differentiation, which correlates with blood flow velocity. Using cell culture they show that pulsatile blood flow promotes pericyte differentiation into smooth muscle cells. They further identify transcription factor klf2a as differentially regulated by blood flow, and show that klf2a inhibition results in VSMC differentiation. The authors conclude that pulsatile flow promotes VSMC differentiation through klf2a activation.

Strengths:

Overall this is an important study, because VSMC differentiation in CW has not been previously studied, although analogous observations regarding the role of blood flow and klf2 involvement have been previously made in other systems and other vascular beds, for example, mouse klf2 mutants, which have deficient VSMC coverage of the dorsal aorta (Wu et al., 2008, JBC 283: 3942-50). The results convincingly show that VSMC differentiation in CW depends on the blood flow, and that klf2a flow dependent function regulates VSMC differentiation.

Weaknesses:

(1) The provided data do not support correlation between wall shear stress (WSS) and acta2+ cell number. The number of acta2+ cells in CaDI increases dramatically between 54 hpf and 3 dpf (Fig. 2F). However, the graph provided in the response to reviewers shows that WSS in CaDI is actually lower at 3 dpf compared to 54 hpf. Authors argue that Pearson correlation analysis shows that both variables increase together, but this is calculated over the stage between 54 hpf and 4 dpf. acta2+ cells appear by 3 dpf, and at this stage WSS in CaDI is not increased (or even lower), which argues agains WSS being the cause of acta2+ cell differentiation. Furthermore, data in Fig. 3I-K show that WSS actually decreases in BCA and PCS between 54 hpf and 4 dpf, while the number of acta2+ increases in BCA and PCS by 4 dpf. This also argues against the argument that WSS affects differentiation of acta2+ cells.

(2) In multiple instances, results are based on a single independent experiment (Fig. 3, Fig. 4H, I, Fig. S2 and Fig. S3) with only a few embryos analyzed in many cases. This falls short of expected standards in the field, and it is unclear if these results are reproducible.

---

## [Author Response]

The following is the authors’ response to the original reviews.

Weaknesses to be addressed:(1) More detail is required to understand the effects of genetic and drug manipulations on heart rate as these are important experiments. At the very least, a discussion on the limitations of these manipulations is needed.- For example, how does one separate the pulsatile versus nutritive effects of blood flow/heartrate reduction?- The conclusion that arterial SMC differentiation is driven by pulsatile blood flow needs to be toned down. Indeed, this conclusion is mainly supported by in vitro cell co-cultures exposed to laminar versus pulsatile flow. In vivo, reducing Tnnt2a expression affects cardiac contractility and blood flow does not selectively affect pulsatility. To make this conclusion, the authors would need an experimental means to selectively dampen the pulsatility of blood flow.

We understand this concern and we toned down the statements related to pulsatile flow of our conclusion by using 'flow' instead of 'pulsatile flow' in all text except for the *in vitro* co-cultures part. We also added a paragraph to discuss the limited capability of qualitatively reduce blood flow *in vivo*, and acknowledge that the effects of nutrients and flow reduction could not be uncoupled in live zebrafish embryos. We proposed that in the future, *in vitro* 3D vascular culture models may be combined with microfluidics to precisely calibrate nutrient composition in culture media, flow velocity and pulse; these methods would help address these questions more thoroughly. See page 11-12 line 312-322.

(2) Since mural cells are sensitive to transmural pressure, could the authors elaborate on the potential role of raised intravascular pressure in SMC differentiation? This would better parallel rodents and humans.

We thank you for this suggestion. We added a paragraph to discuss the potential role of raised intravascular pressure in VSMC differentiation in discussion (see page 11 line 296-311).

(3) The authors use nifedipine to reduce blood flow. Nifedipine is a specific and potent inhibitor of voltage-dependent calcium channels (VDCC) which are expressed in SMCs. Prior studies (PMID: 35588738) showed that VDCC blockers increased rather than inhibited SMC differentiation. Nifedipine is also likely to act upon VSMC calcium handling in the circle of Willis, which may in turn affect cell maturation. Could the authors comment on this seeming discrepancy?It is possible that off-target or indirect effects of Nifedipine decrease smooth muscle cell proliferation, or that altered cardiac contractility fundamentally alters aspects of vascular development other than blood flow.- Additionally, it would be helpful to report the quantitative heart rate reduction achieved with Nifedipine. This would clear up concerns that the heart rate reduction is too large for normal vascular development to occur, and thus decrease proliferation rate independent of changes in blood flow pulsatility.

We concur with these comments, which is why our experiments with nifedipine is complemented by an alternative, non-pharmacological strategy to inhibit blood flow: the use of morpholino against *tnnt2a* gene. The results with either nifedipine or *tnnt2a* MO support lack of VSMC differentiation. In addition, we provided quantitative heart rate reduction achieved with nifedipine shown in new Figure 4-figure supplement1A-C, suggesting that the drug is not completely halting the heart rate but decreasing it. Nevertheless, zebrafish embryos can survive and develop vascular system without heartbeat for several days (Stratman and Weinstein, 2021). Hence, we do not think that the effect on VSMC differentiation is associated with non-specifical effects caused by the decrease of heartbeat. Nevertheless, we now acknowledged in our discussion the limitation of nifedipine, as it may affect VSMC through VDCCs (page 12, line 323-334).

We also added a paragraph in discussion to compare nifedipine, an L-type VDCC blocker, and ML218, a T-type VDCC selective inhibitor from the previous study (Ando et al., 2022). We noted that in this previous study, the increase in VSMC differentiation only occur on anterior metencephalic central arteries (AMCtAs) that are more than 40 mm away from the BCA; these AMCtAs are much smaller than CoW arteries and have different geometry hence possible different kinetics of VSMC differentiation (Ando et al., 2022) as our manuscript discovery would suggest.

(4) The authors should provide more information on how blood flow velocity and wall shear stress are calculated from the Circle of Willis vascular structure. It is presumed that these values are dependent upon the 3-D morphology of the vessel network, as labeled by intravenous dextran dye, but this is not clear. (a second reviewer similarly comments: I was unclear how flow velocity values were obtained in Fig. 3E. Are they based on computational simulation, or are they experimentally calculated following the dextran injection?) Small local differences in vessel diameter and shape will influence blood flow velocity, but these morphological changes are not clearly articulated. Further, it is unclear how flow input levels to the CaDI and basilar arteries are decided across time points. For instance, is it possible to measure the blood flow speed empirically with line-scanning or high-speed tracking of labeled blood cells or particles? This would provide validation of the modeling results.

The computational fluid dynamic simulation was performed according to previous study from our lab (Barak et al., 2021). Blood flow velocity and wall shear stress are dependent on geometry of vessel network labeled by intravascular dextran. More details on how the computational fluid dynamic simulation was performed are added in method section page 17 line 433-449.

Moreover, to address this reviewer concern we have now provided new experimental measurement of blood flow using the red blood cell (RBC) velocity with axial line scanning microscopy in Tg(*kdrl:gfp;gata1:DsRed*)zn1/sd2 zebrafish embryos at 54 hpf, 3 dpf, and 4 dpf. By using the experimental RBC velocity, we re-simulated computational fluid dynamics. The new findings align with our conclusion and are further elaborated upon in response to the reviewer comment listed as point 6. Details on how RBC velocity calculated is added in method section page 16 line 414-431.

(5) Does the cardiac injection of dextran itself affect the diameter of the arteries, given the invasiveness of the procedure? This could be examined in fish with a transgenic endothelial label with and without dextran.

Here, we performed an experiment on wildtype zebrafish at 5 days post fertilization (dpf) with and without Dextran injection, examining the effects of Dextran injection on vessel diameters. As shown in the representative image below, the XZ panel clearly illustrates a Dextran-filled PCS vessel with no alteration in vessel size. Dextran microangiography, a technique employed to obtain vessel geometry with fluorescent microsphere, has been well established in zebrafish (40). Our findings, demonstrating that Dextran does not affect vessel size, are consistent with previous studies utilizing Dextran microangiography.

(6) The data from the microangiography experiment in Figure 3 does not fully support the stated results. The authors report that the CaDI had the highest blood flow speed starting from 54 hpf, but it does not appear to be higher than the other arteries at this time point. Additionally, there is not sufficient evidence that wall shear stress coincides with smooth muscle cell differentiation in the CaDI. Wall shear stress appears to be similar between 54 hpf and 3 dpf in the CaDI, only increasing between 3 dpf and 4 dpf, while differentiation is shown to begin at 3 dpf. The authors need to address this and/or soften conclusions.

First, In response to this specific reviewer concern, we measured red blood cell (RBC) velocity with axial line scanning microscopy to analyze Tg(*kdrl:gfp;gata1:DsRed*)zn1/sd2 zebrafish embryos (the details were added in Method section in the manuscript). We replaced the computational simulated blood flow velocity with RBC velocity in new Figure 3E-G, and re-run the computational simulated wall shear stress (WSS) using the RBC velocity in new Figure 3I-K. We compared RBC velocity and WSS among different vessels at each time point. We confirmed that CaDI has the highest RBC velocity starting from 54 hpf to 4 dpf (new Figure 3A-C, and E-G) and found an overall increase in average WSS from 54 hpf to 4 dpf (new Figure 3A-C, and H). Further, WSS in CaDI was significantly higher than BCA and PCS at 54 hpf, 3 dpf, and 4 dpf (new Figure 3A-C, I-K). Altogether, the CFD simulation suggests that CoW arteries bear different hemodynamic WSS that is associated with spatiotemporal pattern of VSMC differentiation on CoW arteries. (Page 6, line 153-162)

Second, to identify the correlation of WSS and VSMC differentiation in CaDI, we performed Pearson correlation analysis. In the image provided here, we plotted a linear regression between normalized # of *acta2*+ cells in CaDI and WSS with developmental stages (54 hpf, 3 and 4 dpf), and performed Pearson correlation coefficient analysis by using GraphPad Prism 10.0.3. The correlation coefficient r = 0.595, suggesting that the two variables (*acta2*+ cells and WSS) tend to increase together with developmental stages (54 hpf, 3 and 4 dpf).

**Author response image 2. sa3fig2:** 

Third, we softened our conclusion as the RBC velocity across CoW arteries was differentially distributed while VSMC differentiation occurred on these vessels.

(7) It is unclear if acta2 expression is conferring vascular tone, as would be expected if the cells are behaving as mature VSMCs. Does arterial diameter decrease with an increase in acta2 expression? Are acta2-positive mural cells associated with more dynamic changes in arteriole diameter under basal or stimulated conditions?

Thanks for this interesting question. VSMC maturation and its vasoactivity could be further investigated in the future. Our study focused on early stage of VSMC differentiation, in which *pdgfrb*+ progenitors started to express VSMC marker *acta2*. We discussed the onset of *transgelin* expression and loss of *abcc9* expression as markers of VSMC maturation. In addition, a previous study found that VSMC covered vessels in zebrafish brain dilate as early as 4 dpf and constrict at 6 dpf (Bahrami & Childs, 2020). Future study may focus on the association between expression of different VSMC markers and VSMC functional maturation. (page 10, line 272-279)

(8) The authors argue that CoW vessels transition from venous to arterial identity (Fig. 1). However, kdrl is not an ideal arterial marker for this experiment as it is expressed in both arteries and veins. While it is true that many arterial beds have stronger kdrl expression than the veins, its expression in both arteries and veins changes with developmental stage, and its expression level may vary depending on the type of vessel. Therefore, showing that kdrl increases from 32 hpf - 4 dpf in CoW vessels is not convincing because its expression may increase in both venous or arterial vasculature as the vessels mature. In addition, flt4 expression is not exclusively venous; for example, it has noticeable expression in the dorsal aorta at 24-32 hpf stages. It would be helpful to confirm this transition by analyzing additional arterial and venous markers.

We acknowledge this and we added a paragraph to discuss the limitation. We combined loss of *flt4* and increase in *kdrl* to establish the temporal sequence of circle of Willis morphogenesis, arterial specification, and VSMC differentiation. We acknowledged that additional arterial and venous markers need to be analyzed for a more thorough characterization of arterial specification in vertebrate brain vascular development. See page 12 line 335-341.

(9) The authors show that acta2+ VSMCs are absent in tnnt2a MO embryos, concluding that blood flow is required for their differentiation from pericytes. However, there is no data showing that pericytes are still present in tnnt2a MO embryos. Although this has been previously shown by Ando et al 2016, it would be beneficial to confirm in the current study as this is a critical piece of evidence needed for this conclusion.

To determine if blood flow is dispensable for *pdgfrb*+ progenitor recruitment, we performed *tnnt2a* MO (0.35 ng/embryo) injection in Tg(*pdgrb:egfp, kdrl:ras-mcherry*)ncv22/s896. Loss of blood flow did not affect *pdgfrb*+ progenitor emergence around the CoW (new Figure 4-figure supplement1G, H) at 3 days post fertilization (dpf). This is consistent with previous observation in Ando et al 2016 Figure S2C.

(10) The authors show that klf2a MO injected embryos have a reduced number of VSMCs at 3 dpf but a normal number at 4 dpf (Fig. 6), concluding that klf2a is only important to initiate CaDI muscularization. If this is true, it would raise important questions about how VSMCs differentiate at a later stage in the absence of klf2a. For instance, is blood flow not required to differentiate at a later stage, or is there another factor that compensates in the absence of klf2a? The alternative explanation/ caveat is that klf2a MO loses efficacy with development, leading to the recovery of VSMCs at this stage. Therefore, it would be important to confirm this result using a genetic klf2a mutant.

Thank you for pointing this out. We note that based on the *klf2a* reporter line, *klf2a* activity in CoW arterial endothelial cells is highly correlated with the number of *acta2*+ VSMCs in CaDI, BCA and PCS at 3 dpf (r = 0.974, new Figure 5-figure supplement1J). Interestingly however, *klf2a* activity remained stable from 3 dpf to 4 dpf, well beyond initiation of VSMC differentiation. Thus, we speculate sustained *klf2a* expression may support further maturation of VSMCs, as *acta2*+ VSMCs showed distinct morphology at 4 dpf compared with 3 dpf (Figure 2C, D, Page 10, line 268-272). As for the observation that *klf2a* morphants have normal number of VSMCs at 4 dpf, we think that in addition to the temporary effect of morpholino, a proximal explanation is compensation by paralogous *klf2b* in zebrafish. We acknowledge that further characterization of CoW VSMC development in *klf2a* and *klf2b* double genetic mutants (Rasouli et al., 2018; Steed et al., 2016) may help determine whether *klf2b* compensates *klf2a* in CoW VSMC differentiation beyond 4 dpf. See page 10-11 line 292-295.

(11) A large part of the discussion focuses on Notch and Wnt signaling, as downstream Klf2 effectors. While these are reasonable hypotheses to propose, there is no data on the involvement of these pathways in the current study. It seems excessive to speculate on detailed mechanisms of how Klf2 activates Notch and Wnt signaling in the absence of data showing that these pathways are affected in CoW vessels. Therefore, the discussion could be shortened here unless additional data can be obtained to demonstrate the involvement of these pathways in VSMCs in CoW.

We concur and have condensed the discussion on Notch and Wnt signaling as downstream *klf2* effectors.

Minor comments:(1) Line 138 "CaDI is the only vessels in the CoW receiving pulsatile arterial blood low ... ". Adding a reference to support this statement would be useful.

We agree and revised this sentence into ‘CaDI receive proximal arterial feed through lateral dorsal aorta from cardiac outflow tract (Isogai et al., 2001)’. It was also based on our general observation of zebrafish vascular anatomy and blood flow under a confocal microscope.

(2) The image insets in Figs. 1A, 2A, 4E-L, 5A, 6A are quite small. Please make them larger to help the reader interpret the findings.

We agree. We maximized the image size to help the reader interpret the finding, and to visualize confocal images and schematics side-by-side.

(3) The schematics in Figs. 1-2, and 4-6 are helpful, but the different cell types are difficult to see because they are small and their colors/shapes are not very distinct.

We agree. We increased the size and color contrast to provide better visualization of the schematics in new Figures 1-2 and 4-6.

(4) It is stated that there are no diameter differences between different arteries, but statistics are not reported.

The statistics in Figure 3D were performed by two-way analysis of variance (ANOVA) followed by Tukey’s multiple comparisons test, with a single pooled variance. Here we added pairwise comparisons among vessels in the CoW. Hence when not indicated the difference are not significant.

(5) Figure 3F would be better visualized on a log scale, as it is difficult to see the differences between each post-fertilization timepoint.

We agree. In the new Figure 3H, the average wall shear stress (WSS) in CoW arteries is presented on log scale in y axis to see the differences between each post-fertilization timepoint.

(6) Please provide more background and validation on the pericyte cell line, and their use for the questions in this study.

Thank you for the question, TgBAC(*pdgfrb:egfp*)ncv22 was generated and described by Ando et al 2016 to clarify mural cell coverage of vascular endothelium in zebrafish. We added a description in the method section to provide background and validation on this pericyte line (see page 13 line 368-372).

(7) Flow velocity and WSS changes are shown in each vessel in Figs. 3E,G. However, the comparison should be made between different types of vessels to see if there is a statistical difference and PCS, for example, which would explain differences in VSMC coverage.

We agree. We compared the difference among arteries in the CoW at each developmental timepoint and performed ordinary one-way ANOVA with Tukey’s multiple comparisons test. Figure 3E is replaced by new Figure 3E-G and Figure 3G is replaced by new Figure 3I-K.

(8) Similarly, between CaDI, the number of klf2a cells in Fig. 5B should be compared between different vessels, not between different stages of the same vessel.

We agree. In new Figure 5B-E, the number of *klf2a*+ cells per 100 μm vessel length are compared among different vessels at each developmental stage and analyzed by ordinary one-way ANOVA with Tukey’s multiple comparisons test.

(9) When quantifying klf2+ cells in Fig. 5, it would be helpful to quantify klf2 expression level between cells in different vessels. This could be done by quantifying GFP expression in existing images. The difference in expression level may explain the variation between CaDI and PCS more accurately than just the difference in cell number.

The GFP expression reflect the stability of GFP protein and labels discrete nuclei with active *klf2a* transcription. Hence the quantification of GFP level might not give an accurate readout of *klf2a* expression per se but rather of its transcription. For this reason we do not think that this quantification will add to accurate measurement of *klf2a* expression.

(10) Do data points in Figure 4D correspond to different cells in the same chamber experiment? If so, they cannot be treated as independent replicates. Each data point should correspond to an independent replicate experiment.

We agree. Now in the figure legend, we report the number of cells analyzed.

(11) Graph placement is confusing in Figs. 4I, M. An adjacent Fig. 4G shows Nifedipine treated embryos, while the graph next to (Fig. 4I) shows acta+ cell number from tnnt2a 4 dpf experiment. Similarly, the bottom Fig. 4K tnn2a 4 dpf MO experiment has an adjacent graph Fig. 4M, which shows nifedipine treatment quantification, which makes it very confusing.

We agree. We rearranged Figure 4E (representative images of control embryos at 3 dpf and 4 dpf), Figure 4F (*tnnt2a* MO embryos at 3 dpf and 4 dpf), Figure 4G (nifedipine treated embryos at 3 dpf and 4 dpf).

Reference:

Ando, K., Fukuhara, S., Izumi, N., Nakajima, H., Fukui, H., Kelsh, R. N., & Mochizuki, N. (2016). Clarification of mural cell coverage of vascular endothelial cells by live imaging of zebrafish. Development, 143(8), 1328-1339. https://doi.org/10.1242/dev.132654

Ando, K., Tong, L., Peng, D., Vazquez-Liebanas, E., Chiyoda, H., He, L., Liu, J., Kawakami, K., Mochizuki, N., Fukuhara, S., Grutzendler, J., & Betsholtz, C. (2022). KCNJ8/ABCC9-containing K-ATP channel modulates brain vascular smooth muscle development and neurovascular coupling. Dev Cell, 57(11), 1383-1399 e1387. https://doi.org/10.1016/j.devcel.2022.04.019

Bahrami, N., & Childs, S. J. (2020). Development of vascular regulation in the zebrafish embryo. Development, 147(10). https://doi.org/10.1242/dev.183061

Barak, T., Ristori, E., Ercan-Sencicek, A. G., Miyagishima, D. F., Nelson-Williams, C., Dong, W., Jin, S. C., Prendergast, A., Armero, W., Henegariu, O., Erson-Omay, E. Z., Harmanci, A. S., Guy, M., Gultekin, B., Kilic, D., Rai, D. K., Goc, N., Aguilera, S. M., Gulez, B., . . . Gunel, M. (2021). PPIL4 is essential for brain angiogenesis and implicated in intracranial aneurysms in humans. Nat Med, 27(12), 2165-2175. https://doi.org/10.1038/s41591-021-01572-7

Isogai, S., Horiguchi, M., & Weinstein, B. M. (2001). The vascular anatomy of the developing zebrafish: an atlas of embryonic and early larval development. Dev Biol, 230(2), 278-301. https://doi.org/10.1006/dbio.2000.9995

Kamei, M., Isogai, S., Pan, W., & Weinstein, B. M. (2010). Imaging blood vessels in the zebrafish. In Methods in cell biology (Vol. 100, pp. 27-54). Elsevier.

Rasouli, S. J., El-Brolosy, M., Tsedeke, A. T., Bensimon-Brito, A., Ghanbari, P., Maischein, H. M., Kuenne, C., & Stainier, D. Y. (2018). The flow responsive transcription factor Klf2 is required for myocardial wall integrity by modulating Fgf signaling. Elife, 7. https://doi.org/10.7554/eLife.38889

Steed, E., Faggianelli, N., Roth, S., Ramspacher, C., Concordet, J. P., & Vermot, J. (2016). klf2a couples mechanotransduction and zebrafish valve morphogenesis through fibronectin synthesis. Nat Commun, 7, 11646. https://doi.org/10.1038/ncomms11646